# SONICS: Synthetic Or Not - Identifying Counterfeit Songs

**Md Awsafur Rahman**\*
UC Santa Barbara, USA
awsaf@ucsb.edu

**Zaber Ibn Abdul Hakim**,\* **Najibul Haque Sarker**\*
Virginia Tech, USA
{zaberhakim666, najibulhaque}@vt.edu

**Bishmoy Paul**\*
Santa Clara University, USA
bpaul@scu.edu

**Shaikh Anowarul Fattah**
BUET, Bangladesh
fattah@eee.buet.ac.bd

## ABSTRACT

The recent surge in AI-generated songs presents exciting possibilities and challenges. These innovations necessitate the ability to distinguish between human-composed and synthetic songs to safeguard artistic integrity and protect human musical artistry. Existing research and datasets in fake song detection only focus on singing voice deepfake detection (SVDD), where the vocals are AI-generated but the instrumental music is sourced from real songs. However, these approaches are inadequate for detecting contemporary end-to-end artificial songs where all components (vocals, music, lyrics, and style) could be AI-generated. Additionally, existing datasets lack music-lyrics diversity, long-duration songs, and open-access fake songs. To address these gaps, we introduce SONICS[1], a novel dataset for end-to-end Synthetic Song Detection (SSD), comprising over 97k songs (4,751 hours) with over 49k synthetic songs from popular platforms like Suno and Udio. Furthermore, we highlight the importance of modeling long-range temporal dependencies in songs for effective authenticity detection, an aspect entirely overlooked in existing methods. To utilize long-range patterns, we introduce SpecTT-Tra, a novel architecture that significantly improves time and memory efficiency over conventional CNN and Transformer-based models. For long songs, our top-performing variant outperforms ViT by 8% in F1 score, is 38% faster, and uses 26% less memory, while also surpassing ConvNeXt with a 1% F1 score gain, 20% speed boost, and 67% memory reduction.

## 1 INTRODUCTION

The rapid advancements in AI-generated music present a substantial threat to the music industry, potentially reducing the demand for professional musicians and stifling new talent development (McMahon, 2024; Derbyshire et al., 2023). To preserve the unique value of human creativity, it is crucial to develop robust methods for detecting AI-generated music, ensuring a fair and vibrant creative ecosystem.

Singing Voice Synthesis (SVS) (Liu et al., 2022a) and Singing Voice Conversion (SVC) (Jayashankar et al., 2023) have recently achieved significant progress, enabling the creation of synthetic singing voices that closely mimic real singers' styles. When combined with instrumental music from real songs, these synthetic voices can produce convincing counterfeit songs. Although related to synthetic speech detection, detecting fake songs is particularly challenging due to the unique rhythmic patterns and artistic vocal traits of singing (Zang et al., 2024b). To address this, researchers have turned their attention to Singing Voice Deepfake Detection (SVDD) (Xie et al., 2024; Zang et al., 2024b;a). However, current methods relying on datasets composed of SVS and SVC-generated songs face several limitations. These datasets are bound to use only instrumental music from real songs, leading to artifacts like the "*Karaoke effect*" (volume discrepancies

---

\*Equal contribution.
[1]Code & Data available at https://github.com/awsaf49/sonics

Table 1: Comparison of Proposed and Existing Fake Song Datasets

| Dataset | End-To-End Fake Songs | Text Lyrics | Song Style | Music-Lyrics Diversity | Open Fake Songs | Open Real Songs | Open-Source Models | Language | Average Length (sec) | # Algorithms | # Speakers | # Real Songs | # Fake Songs | # Total Songs | # Total Hours |
|---|---|---|---|---|---|---|---|---|---|---|---|---|---|---|---|
| FSD (Xie et al., 2024) | | | | | ✓ | ✓ | ✓ | Chinese | 216.00 | 5 | 60 | 200 | 450 | 650 | 26 |
| SingFake (Zang et al., 2024b) | | | | | | | | Multi | 13.75 | - | 40 | 634 | 671 | 1,305 | 58 |
| CtrSVDD (Zang et al., 2024a) | | | | | ✓ | ✓ | ✓ | Multi (no English) | 4.87 | 14 | 164 | 32,312 | 188,486 | 220,798 | 307 |
| SONICS (ours) | ✓ | ✓ | ✓ | ✓ | ✓ | | | English | 176.03 | 5 | 9,096+ | 48,090 | 49,074 | 97,164 | 4,751 |

between music and vocals) and limited music-lyrics diversity. Moreover, existing methods overlook the long-context temporal relationships inherent in songs, such as repeated verses, music, rhythm, and emotional dynamics, which are critical for effective detection. Availability of only short duration songs in current datasets further hampers the use of these patterns. Additionally, copyright restrictions on some existing datasets limit the public availability of generated fake songs, hindering broader usage. Furthermore, the SVDD task requires separate tools for voice identification and separation during data processing (Xie et al., 2024; Zang et al., 2024b), increasing computational overhead.

Recently, platforms like Suno[2] and Udio[3] have gained significant traction on social media. They can synthesize not only vocals but also entire songs, including synthetic music, styles, and lyrics, further complicating the situation. Due to their end-to-end nature, these fake songs differ significantly from those generated by SVS and SVC methods, rendering existing SVDD methods and datasets inadequate for detecting them. This necessitates an urgent need for a detection system specifically designed for end-to-end synthetic song detection (SSD).

To address these shortcomings, we introduce SONICS, a large-scale dataset comprising 97,164 songs (4,751 hours), including 49,074 end-to-end synthetic songs (1,971 hours) generated by Suno and Udio, alongside 48,090 real songs (2,780 hours) curated from YouTube. With an average duration of 176 second (sec.), the SONICS dataset supports the use of long-context relationships in songs for accurate fake song detection. Furthermore, SONICS addresses the issue of music-lyrics diversity by including a wide range of music styles and both real and synthetic lyrics. A unique feature of SONICS is that it includes the text lyrics of songs, which can aid future research.

Despite the availability of long songs in our dataset, utilizing long-context relationships presents additional challenges. For instance, CNN-based models struggle to capture long-range dependencies due to their inherently local receptive fields. While Transformer-based models can capture these dependencies with global attention, they are computationally expensive with longer audio inputs. To mitigate this trade-off, we introduce **Spec**tro-**T**emporal **T**okens **Tra**nsformer (**SpecTTTra**), which uses a Spectro-Temporal Tokenizer to significantly reduce computational costs while employing global attention. Our contributions are summarized as follows:

- We introduce SONICS, the first large-scale dataset for end-to-end synthetic song detection that addresses the limitations of existing datasets, including limited music-lyrics diversity, short-duration songs, and open fake songs.

- We provide a human benchmark for fake song detection, filling a gap in previous work, and establish a standard AI benchmark for CNN and Transformer-based models.

- We highlight the importance of modeling long-context temporal relationships in songs, an aspect entirely overlooked in existing approaches.

- We propose a faster and memory efficient model, SpecTTTra, which effectively captures long-context temporal relationships in songs while outperforming popular methods.

## 2 RELATED WORKS

**Synthetic Speech Detection:** The domain of synthetic speech detection, closely tied to synthetic song detection through their shared audio modality, has been extensively explored due to advancements in voice conversion (Zhao et al., 2020) and synthesis techniques (Wang et al., 2021). These

---

[2]https://suno.com, 2022. Accessed: 2024-06-27
[3]https://udio.com, 2023. Accessed: 2024-07-09

Table 2: Performance of SingFake-trained models on SingFake vs SONICS dataset

| Model | SingFake | | | | SONICS | | | |
|---|---|---|---|---|---|---|---|---|
| | EER ↓ | F1 ↑ | Sens. ↑ | Spec. ↑ | EER ↓ | F1 ↑ | Sens. ↑ | Spec. ↑ |
| ConvNeXt | 0.20 | 0.86 | 0.90 | 0.65 | 0.38 | 0.33 | 0.22 | 0.88 |
| ViT | 0.29 | 0.84 | 0.97 | 0.35 | 0.49 | 0.64 | 0.87 | 0.16 |
| EfficientViT | 0.19 | 0.88 | 0.93 | 0.64 | 0.50 | 0.35 | 0.28 | 0.71 |

advancements have spurred the development of audio spoofing attacks on speaker verification systems and deepfake audio targeting human listeners (Kawa et al., 2023). Synthetic speech detection methods include Light CNN (LCNN) with Max-Feature-Map activations (Lavrentyeva et al., 2019), Transformer encoders with ResNet architectures (Zhang et al., 2021b), RawNet2 with sinc layers and GRU blocks (Tak et al., 2021), and heterogeneous graph attention networks (Jung et al., 2022). However, the unique complexities of songs—such as rhythm, melody, and emotional nuance—present challenges that traditional speech detection methods are not equipped to handle, as shown by Xie et al. (2024) and Zang et al. (2024b). Thus, following CtrSVDD (Zang et al., 2024a), we opted not to conduct similar experiments in our study.

**Synthetic Song Detection:** Synthetic song detection, a relatively newer and more complex challenge, has gained attention recently. In early 2024, SingFake (Zang et al., 2024b) introduced a dataset of counterfeit songs using Singing Voice Conversion (SVC), along with the task of Singing Voice Deepfake Detection (SVDD) and associated model benchmarks. Subsequent work (Xie et al., 2024; Zang et al., 2024a) combined Singing Voice Synthesis (SVS) with SVC to create phoneme-based songs, leading to specialized detection datasets. Methods in this area include convolutional networks for feature extraction followed by classification using graph neural network (Jung et al., 2022), wav2vec2-based extraction coupled with graph neural networks (Tak et al., 2022), Linear-Frequency Cepstral Coefficients (LFCC) used with ResNet18 models (Zhang et al., 2021a) and combination of music-specific models (MERT) & linguistic models (wav2vec2.0) with targeted augmentations (Chen et al., 2024). However, SVC and SVS-based datasets retain original background music, leading to a detectable "*Karaoke effect*" artifact. Recent end-to-end fake songs by Suno and Udio, can produce divergent fake songs where all musical components (e.g., background music, styles, and lyrics) can be synthetic, presenting a severe detection challenge. As shown in Table 2, models trained on SingFake dataset, perform poorly when tested on our end-to-end fake songs dataset, with a significant drop in detection performance (F1 score) ranging from 20% to 64%.

**Long Audio Classification:** Songs exhibit long-range temporal patterns, such as repeated verses, rhythms, etc. setting them apart from speech (Albouy et al., 2024). Despite their potential to enhance detection performance, these patterns have been largely overlooked in existing methods (Xie et al., 2024; Zang et al., 2024b;a). Meanwhile, long audio classification remains a relatively less explored area in audio research. Although automatic speech recognition handles long audio data (Koluguri et al., 2024), it struggles with end-to-end processing of extended audio due to its high computational cost, thus often uses sliding window techniques (Gulati et al., 2020; Radford et al., 2023) to manage costs. This further complicates leveraging long-context features for fake song detection.

## 3 METHODOLOGY

### 3.1 SONICS DATASET

The development of a modern synthetic song detection system necessitates a dataset that meets several stringent criteria, which are conspicuously absent in existing music datasets. These criteria include: **1)** songs where all components—not just vocals—can be AI-generated; **2)** song lengths sufficient to capture long-term contextual relationships; **3)** a diverse spectrum of music-lyrics combinations; and **4)** a quantity of data substantial enough to serve as a generative model benchmark. Addressing these needs, we introduce the SONICS dataset, a comprehensive collection of end-to-end AI-generated songs produced using the latest audio generative models, spanning lengths from 32 to 240 sec and encompassing an extensive array of music-lyrics styles. A detailed comparison of SONICS with existing datasets is presented in Table 1. It clearly illustrates that datasets such as FSD (Xie et al., 2024), SingFake (Zang et al., 2024b), and CtrSVDD (Zang et al., 2024a) fall short of fulfilling all the outlined criteria when juxtaposed with SONICS. Additionally, a comprehensive distribution summary of the SONICS dataset is provided in Table 3.

Table 3: Summary of our proposed SONICS dataset

| Split | Full Fake | | | | | Half Fake | | | Mostly Fake | | | | | Real | Total |
|---|---|---|---|---|---|---|---|---|---|---|---|---|---|---|---|
| | Suno v3.5 | Udio 130 | Suno v3 | Suno v2 | Udio 32 | Suno v3.5 | Suno v3 | Suno v2 | Suno v3.5 | Udio 130 | Suno v3 | Suno v2 | Udio 32 | YouTube | |
| # Train | 774 | 656 | 0 | 0 | 0 | 4558 | 0 | 0 | 11863 | 16172 | 0 | 0 | 0 | 32686 | 66709 |
| # Test | 77 | 53 | 206 | 100 | 203 | 396 | 634 | 309 | 1117 | 1566 | 2797 | 1350 | 3970 | 13237 | 26015 |
| # Valid | 9 | 12 | 37 | 16 | 30 | 67 | 117 | 51 | 196 | 286 | 494 | 258 | 700 | 2167 | 4440 |

### 3.1.1 REAL SONGS

The dataset's Real songs segment comprises original compositions by human artists. This portion of the dataset was initially compiled by random sampling from the Genius Lyrics Dataset (J., 2023), which provides metadata including lyrics, titles, and artist information. Subsequently, leveraging this metadata, a search was conducted dynamically on YouTube to retrieve the corresponding audio files. This process yielded 48,090 songs performed by 9,096 artists.

### 3.1.2 FAKE SONGS

For generating end-to-end fake songs, we utilized the only two currently available audio generative model families: Suno and Udio. Specifically, we used three variants from Suno—Suno v2, v3, and v3.5. Suno v2, the earliest iteration, generates songs up to 80 seconds (sec.) in length. Suno v3 extends this capability to 120 sec. The latest iteration, Suno v3.5, produces songs up to 240 sec. From the Udio family, we used Udio 32 and Udio 130, capable of generating songs of 32 and 130 sec, respectively. For generating and downloading songs from Suno, we utilized the community API (gcui art, 2024). In contrast, for Udio, we manually generated the songs from the website and then downloaded them using community API (Riera, 2024). Subsequently, we used PyAnnote (Bredin & Laurent, 2021) to filter out audio files lacking vocal components, which were predominantly found among the Udio-generated song.

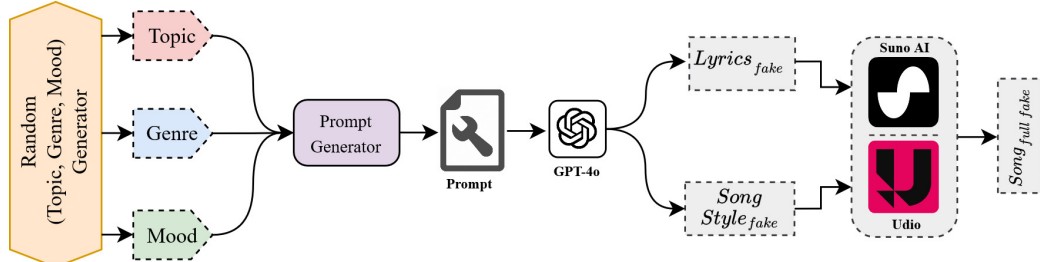

Figure 1: End-to-end pipeline of Full Fake song generation process. Here a random combination of topic, genre, and mood is utilized to create fake lyrics and fake song styles by prompt engineering LLM GPT-4o. The generated lyrics and styles are utilized to create synthetic songs through audio-generative models from Suno and Udio.

The Fake songs are generated with Suno and Udio models by two primary text inputs: **1)** lyrics and **2)** song-style (e.g., instruments, genre, vocal type) where for each combination of (lyrics, style) input, two songs are generated. Based on the nature of these inputs, the Fake songs are categorized into three distinct groups: **i)** Full Fake (FF)—featuring both AI-generated lyrics and song-style; **ii)** Mostly Fake (MF)—where lyrics are AI-generated based on real lyrics features and song-style is derived from real songs; and **iii)** Half Fake (HF)—where the lyrics are directly sourced from real songs, with song-style also extracted from real songs. To generate FF songs, we curated a rich set of metadata, including 57 broad topics (e.g., friendship, betrayal), 292 specific topics (e.g., star trek, pokemon), 49 music genres (e.g., rock, metal), and 72 moods (e.g., calm, angry). The full list can be found in the Appendix. Random combinations of these elements were used to generate final lyrics and styles via a Large Language Model (LLM). The complete pipeline for FF song generation is illustrated in Fig. 1. Here the prompt generator generates a prompt for later stages by filling in the variable values into the placeholders of a predefined prompt template (details in Appendix H). In

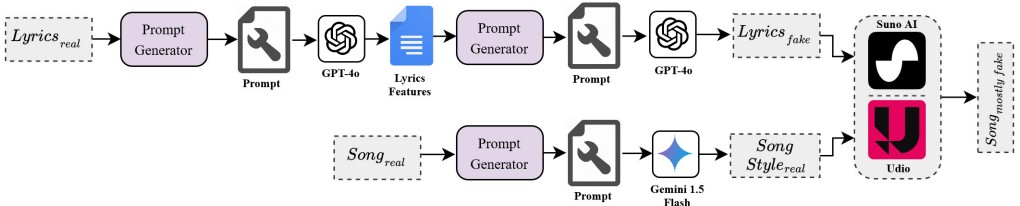

Figure 2: End-to-end pipeline of Mostly Fake song generation process. Firstly, lyrics features are extracted using an LLM from real song lyrics, which in turn is utilized to create fake lyrics through a second call to the LLM GPT-4o. Secondly, style information is extracted from real songs using multimodal Generative model Gemini 1.5 Flash. Finally, the generated lyrics and styles are utilized to create synthetic songs through audio-generative models from Suno and Udio.

contrast, the creation of MF and HF songs involved extracting song-styles from real songs using the multimodal LLM Gemini 1.5 Flash, which uniquely supports audio input. Additionally, for MF songs, lyrics were generated using the GPT-4o LLM, which extracts features from existing real song lyrics to produce new lyrics with a similar distribution, avoiding direct replication of existing content. The pipeline for generating MF songs is detailed in Fig. 2. The HF songs, differing only in the use of real lyrics directly sourced from existing songs, follow a similar process as MF songs. It is noteworthy that, Udio models cannot generate songs for lyrics from real songs; thus, HF songs contain only Suno models. In total, we generated 6,132 HF songs, 40,769 MF songs, and 2,173 FF songs using Suno and Udio models. The prompt templates used in the LLM to extract lyrics features and song styles, as well as to generate lyrics and styles, can be found in the Appendix.

## 3.2 MODEL

The architecture of an audio classification model depends on how the input audio is processed for feature extraction. In our benchmark, we use Mel-spectrogram due to its versatile usage and effective performance across various audio processing tasks (Gong et al., 2021; Radford et al., 2023; Rouard et al., 2023; Niizumi et al., 2024; Shul & Choi, 2024). Spectrograms, resembling 2D shape of an image, allow the usage of image classification models for audio classification. Since image classification models are generally categorized into CNN-based and Transformer-based architectures, we use popular models from both categories in our study, such as ConvNeXt (Liu et al., 2022b) and ViT (Dosovitskiy et al., 2020), for benchmarking. Additionally, we employ EfficientViT (Cai et al., 2023), a hybrid model that integrates both Transformer and CNN components. However, these models encounter challenges when dealing with the inherent long-range temporal dependencies in songs. Specifically, ConvNeXt struggles to capture long-range dependencies due to its local receptive field, a characteristic inherited from CNNs. On the other hand, while ViT is capable of capturing long-range dependencies, it incurs significant computational costs as the number of patches/tokens rapidly increases with longer audio inputs. Similarly, EfficientViT, despite its linear global attention, becomes computationally expensive for long audio due to the large number of tokens combined with its multi-scale operations. To address this issue, we introduce SpecTTTra, which utilizes global attention similar to ViT but employs a Spectro-Temporal Tokenizer to reduce the number of tokens considerably, thereby lowering computational costs and enhancing efficiency significantly.

### 3.2.1 SPECTRO-TEMPORAL TOKENS TRANSFORMER (SPECTTTRA)

As illustrated in Fig. 3, the proposed SpecTTTra model begins by applying temporal and spectral slicing to copies of the input spectrogram, generating distinct temporal and spectral clips/patches indicated by $|t_i|$ and $|f_i|$ respectively where $i$ denotes the position of each clip. These clips are then processed by the Spectro-Temporal Tokenizer, where temporal clips are embedded as temporal tokens $\langle t_i \rangle$ and spectral clips as spectral tokens $\langle f_i \rangle$, using separate tokenizers, where each token is represented as a vector of shape $(1, embed\_dim)$. Subsequently, separate positional embeddings ($p_i^t$ for temporal and $p_i^f$ for spectral) are added to the respective tokens. Separate embeddings are used because there is no positional relationship between temporal and spectral tokens.

The positionally aware tokens are then fed into a ViT-like Transformer encoder, where they are contextualized with one another through global attention. This process enables the temporal tokens

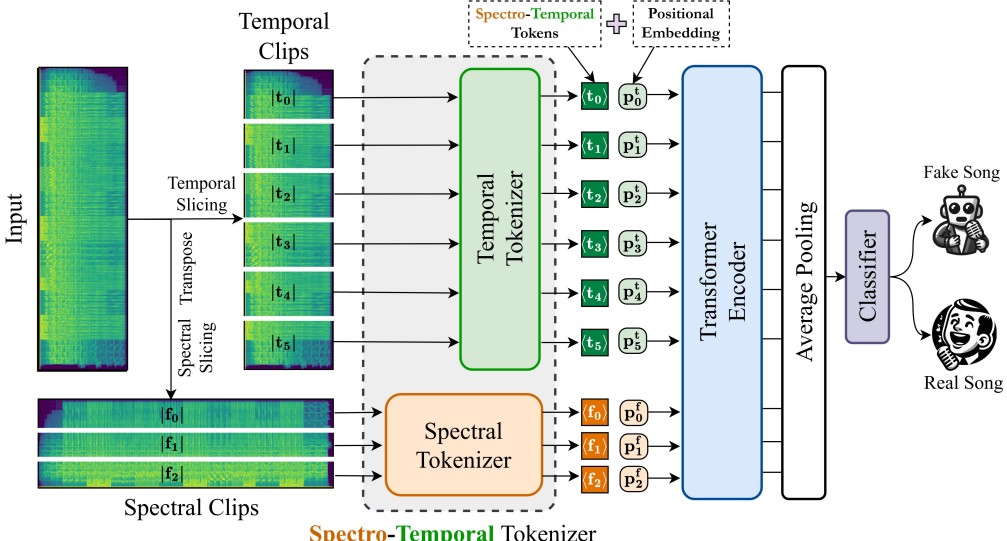

Figure 3: Proposed Spectro-Temporal Tokens Transformer (SpecTTTra) model. First, the input mel-spectrogram undergoes separate temporal and spectral slicing to generate corresponding clips, which are tokenized into temporal and spectral tokens using separate tokenizers. Next, separate positional embeddings are added to these tokens, which are then passed through a Transformer encoder. The resulting globally contextualized features are then average pooled and finally passed to the classifier.

to become aware of both other temporal tokens and spectral tokens, and vice versa for spectral tokens. These globally aware features, now of shape $(n\_tokens, embed\_dim)$, are average-pooled across the $n\_tokens$ dimension to aggregate the temporal and spectral information. Finally, the accumulated features are passed to a classifier, where they are classified as either real or fake songs.

It is important to note that while previous work has attempted to utilize both spectral and temporal information, these approaches come with significant limitations. For instance, Yadav et al. (2023); Gong et al. (2021) focuses solely on temporal tokens, neglecting the rich spectral information. On the other hand, methods like Zadeh et al. (2019); Shul & Choi (2024) employ separate attentions for spectral and temporal information but with ViT-like tokens, which, as discussed in the following section, become highly inefficient and computationally expensive as they rapidly increase with longer audio inputs. In contrast, our approach disentangles spectral and temporal information at the tokenization level and later contextualizes them with attention, leading to greater efficiency.

### 3.2.2 SPECTRO-TEMPORAL TOKENIZATION

The ViT model creates grids of small square patches by simultaneously dividing a 2D spectrogram along both temporal and spectral dimensions, resulting in each patch having access to only limited spectral and temporal information. More critically, as indicated in Eq. 1, the number of patches/tokens ($N_\nu$) increases rapidly with the temporal dimension $T$. For example, for a spectrogram of size $F = 128$ (spectral dimension) with a short audio duration of 5 sec ($T = 128$) , ViT (patch size, $p = 16$) generates tokens $N_\nu = 64$. However, for longer audio with 120 sec ($T = 3744$), the number of tokens surges to $N_\nu = 1872$, nearly 30x more than for the shorter audio. Since the computational cost for global attention in ViT scales quadratically with the number of tokens, this makes it less practical for long audio classification. Similarly, EfficientViT, despite having linear global attention, multi-scale operations coupled with numerous tokens still result in high complexity.

In contrast, SpecTTTra performs slicing independently for temporal and spectral dimensions, as shown in Fig. 3. This approach ensures that each temporal patch has access to the full spectral information, and each spectral patch contains the complete temporal information. This design leverages the observation that meaningful correlations can exist between distant temporal clips (e.g., between $|t_0|$ and $|t_4|$, capturing repeated song verses) or between distinct spectral clips (e.g., between $|f_0|$ and $|f_2|$, capturing harmonics). Importantly, as demonstrated in Eq. 2, due to the additive nature of token generation in SpecTTTra, the number of tokens grows more slowly compared to ViTs, signif-

| **Equation for ViT Token Count** |
| If $T$ and $F$ denote the temporal and spectral dimensions of the spectrogram, respectively, and $p$ is the patch size, then the number of patches (or tokens) generated by a ViT ($N_\nu$) can be expressed as: |

$$N_\nu = \left(\frac{F}{p}\right) \times \left(\frac{T}{p}\right) \qquad (1)$$

| **Equation for SpecTTTra Token Count** |
| If $t$ and $f$ represent the sizes of the temporal and spectral clips, respectively, then the number of patches (or tokens) generated by SpecTTTra, denoted as $N_\psi$, is given by: |

$$N_\psi = N_{spectral} + N_{temporal}$$
$$= \left(\frac{F}{f}\right) + \left(\frac{T}{t}\right) \qquad (2)$$

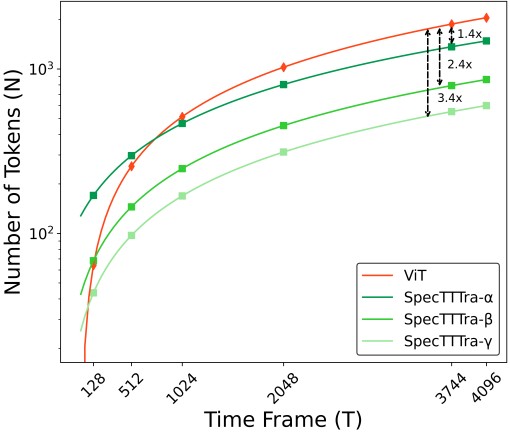

Figure 4: Comparison of the number of tokens generated by the ViT model and the three SpecTTTra variants ($\alpha$, $\beta$, and $\gamma$) as a function of the number of time frames.

icantly reducing computational costs. For instance, using the same parameters as before and setting the temporal and spectral clip sizes to $t = 7$ and $f = 5$, the number of tokens in SpecTTTra for short audio is $N_\psi = 43$, and for long audio, it increases to $N_\psi = 560$, which is approximately 3.4x fewer than in the ViT model. This substantial reduction in token count makes SpecTTTra significantly more computationally efficient. We further explore three variants of SpecTTTra, differentiated by the sizes of their temporal ($t$) and spectral ($f$) patches: SpecTTTra-$\alpha$ ($f = 1$, $t = 3$), SpecTTTra-$\beta$ ($f = 3$, $t = 5$), and SpecTTTra-$\gamma$ ($f = 5$, $t = 7$). Fig. 4 illustrates the rate at which the number of tokens increases for different SpecTTTra variants and ViT as the audio length grows.

The resulting spectral and temporal clips are processed by the Spectro-Temporal Tokenizer (STT) to create spectral and temporal tokens, respectively. The proposed STT block consists of separate Spectral and Temporal Tokenizers, which share identical architecture but differ in their objectives based on their input. These tokenizers use a Linear layer to map the spectral/temporal clips to spectral/temporal tokens, followed by GELU activation, addition of learnable positional embeddings similar to ViT, and finally, layer normalization. For efficiency, during implementation, the slicing and tokenization operations are merged using 1D CNN layer. Given an input spectrogram $x \in \mathbb{R}^{F \times T}$, the mathematical expression for tokenization is as follows:

$$
\begin{aligned}
x_t &= \text{Conv1D}\left(x, c_i = F, c_o = D, k = t, s = t\right) \\
\langle t \rangle &= \text{LayerNorm}\left(\text{GELU}\left(x_t^\top\right) + \hat{p}^t\right) \\
x_f &= \text{Conv1D}\left(x^\top, c_i = T, c_o = D, k = f, s = f\right) \\
\langle f \rangle &= \text{LayerNorm}\left(\text{GELU}\left(x_f^\top\right) + \hat{p}^f\right)
\end{aligned}
\qquad (3)
$$

Here, $\langle t \rangle \in \mathbb{R}^{\frac{T}{t} \times D}$ and $\langle f \rangle \in \mathbb{R}^{\frac{F}{f} \times D}$ denote the temporal and spectral tokens, where $D$ is the token embedding dimension. The parameters $c_i$, $c_o$, $k$, and $s$ represent the input channels, output channels, kernel size, and stride size of the CNN layers. Additionally, $\hat{p}$ and $\top$ denote the learnable positional embeddings and Transpose operation. Further implementation details can be found in the Appendix.

## 4 EXPERIMENTS AND RESULTS

### 4.1 DATASET

We conduct all experiments using the proposed SONICS dataset, which is divided into train, valid, and test sets. To ensure comprehensive evaluation, the valid and test sets include cases with unseen algorithms (e.g., Suno v2, Suno v3, Udio 32) and unseen singers. We also prevent data leakage by ensuring that song pairs from the same (lyrics, style) inputs are exclusively in either the training or valid-test sets, not in both. The distribution of the train, test, and valid sets is shown in Table 3.

## 4.2 IMPLEMENTATION DETAILS

To train models, we resampled both real and fake songs to 16kHz and generated spectrograms with n_fft = win_length = 2048, hop_length = 512, and n_mels = 128, yielding a $128 \times 128$ spectrogram for 5 sec and $128 \times 3744$ for 120 sec audio. Any song shorter than input length is zero-padded randomly, while for longer songs, a random crop is used. We also apply MixUp (Zhang, 2017) and SpecAugment (Park et al., 2019) augmentations during training to improve generalization. However, during the test, to maintain determinism, padding is done on the right side and cropped segments are taken from the middle. We conduct our training on an NVIDIA A6000 GPU with 48GB RAM, using WandB for tracking. We use ViT-small (patch size = 16) and ConvNeXt-tiny along with EfficientViT-B2 from the timm (Wightman, 2019) library. In SpecTTTra, we use the same model configuration as ViT-small. We train all models for 50 epochs from scratch using Binary Cross-Entropy loss with 0.02 label smoothing (Szegedy et al., 2016). Optimization is performed with AdamW (Loshchilov, 2017) and a cosine learning rate scheduler from timm, including a 5-epoch warm-up. While existing methods (Zang et al., 2024b;a; Xie et al., 2024) use Equal Error Rate (EER) as a metric, we prioritize the F1 score (binary average, threshold = 0.5) as our primary metric due to EER's susceptibility to class imbalance. We also evaluate Sensitivity (Sens.) and Specificity (Spec.) to assess performance across fake and real classes.

## 4.3 BENCHMARKS

### 4.3.1 AI BENCHMARK

The comparative analysis of the proposed SpecTTTra models against other existing models is presented in Table 4. The results reveal a significant performance gain (6% for ConvNeXt, 8% for EfficientViT, 10% for ViT, and 17% for SpecTTTra-$\alpha$) in the overall F1 score when using long songs. This finding substantiates our claim that leveraging long-context information is crucial for enhancing fake song detection. Additionally, the advantage of longer audio duration is more prevalent in transformer-based models such as ViT and SpecTTTra, as well as in the hybrid EfficientViT model, compared to the CNN-based ConvNeXt. Notably, the proposed SpecTTTra-$\alpha$, while trailing ConvNeXt by 10% in the F1 score for short audio, outperforms it in long audio. This can be attributed to the global attention mechanism in transformer models, which effectively captures long-range dependencies within the input data. In contrast, models with CNN components tend to perform better on shorter audio. Specifically, ConvNeXt and EfficientViT achieve overall F1 scores of 90% and 87%, respectively, outperforming all transformer-based models on short audio. However, despite the absence of global attention, ConvNeXt demonstrates competitive performance compared to SpecTTTra-$\alpha$ on long audio and outperforms ViT, EfficientViT, and other SpecTTTra variants in both short and long audio scenarios. We hypothesize that this is due to the inherent inductive biases present in CNNs, which are lacking in transformers, leading the latter to require larger datasets to reach their true potential (Liu et al., 2021; Dosovitskiy et al., 2020). Another intriguing observation is the performance of ViT, which, despite its large number of tokens (or patches), is outperformed by the $\alpha$ and $\beta$ variants of SpecTTTra and is only on par with the $\gamma$ variant in terms of overall F1 score for long audio, reinforcing SpecTTTra's effectiveness. We hypothesize that this is due to an overload of redundant information from ViT's numerous patches, which may not contribute effectively to the detection task. Moreover, it can also be observed across all models that real songs are more easily identified than fake ones, as indicated by higher specificity and lower sensitivity scores.

Diving deeper into different partitions of test data, we observe that all detection models achieve better performance on seen algorithms (Suno v3.5 and Udio 130) compared to unseen ones (Suno v2, Suno v3, and Udio 32). Particularly, they struggle more with the Udio algorithms, with the most pronounced difficulty observed for Udio 32. However, ConvNeXt and SpecTTTra-$\alpha$ perform relatively well in detecting the Udio 32 algorithm, achieving a sensitivity of 96% and 95% respectively. Interestingly, despite being an unseen algorithm, the detectors perform comparably well on Suno v3 as they do on the seen Suno v3.5 algorithm, suggesting a possible algorithmic similarity between the two. Conversely, for short audio samples, the detectors perform slightly better on songs with seen speakers than those with unseen speakers, a gap that diminishes when longer audio is used. Finally, in Fake Type partitions, all detectors excel in identifying HF songs, likely due to the exclusive presence of Suno algorithms, where detectors generally perform better compared to Udio algorithms. Among MF and FF songs, the models exhibit a slightly lower performance pattern on FF songs.

Table 4: Performance comparison of SpecTTTra and conventional AI models on varying audio lengths, with F1 score as the primary evaluation metric. Here Real/Human and Fake/AI songs denoting Negative and Positive classes, respectively. † indicates unseen algorithms during training.

| Len. (sec) | Model | Algorithm (Sens.) | | | | | Singer (Spec.) | | Fake Type (Sens.) | | | Overall | | |
|---|---|---|---|---|---|---|---|---|---|---|---|---|---|---|
| | | Suno† v2 | Suno† v3 | Suno v3.5 | Udio† 32 | Udio 130 | Seen | Unseen | Half Fake | Mostly Fake | Full Fake | F1 | Sens. | Spec. |
| 5 | ConvNeXt | 0.62 | 0.99 | 0.99 | 0.62 | 0.99 | 0.99 | 0.99 | 0.90 | 0.82 | 0.80 | **0.90** | 0.82 | 0.98 |
| | ViT | 0.79 | 0.95 | 0.98 | 0.57 | 0.86 | 0.79 | 0.79 | 0.92 | 0.78 | 0.76 | 0.79 | 0.80 | 0.79 |
| | EfficientViT | 0.66 | 0.98 | 0.99 | 0.49 | 0.97 | 0.99 | 0.98 | 0.90 | 0.76 | 0.74 | 0.87 | 0.78 | 0.98 |
| | SpecTTTra-$\gamma$ | 0.51 | 0.98 | 0.99 | 0.10 | 0.99 | 0.98 | 0.97 | 0.87 | 0.61 | 0.62 | 0.76 | 0.63 | 0.98 |
| | SpecTTTra-$\beta$ | 0.61 | 0.98 | 0.99 | 0.18 | 0.99 | 0.95 | 0.94 | 0.89 | 0.66 | 0.66 | 0.78 | 0.69 | 0.94 |
| | SpecTTTra-$\alpha$ | 0.68 | 0.99 | 0.99 | 0.26 | 0.99 | 0.93 | 0.92 | 0.91 | 0.69 | 0.70 | 0.80 | 0.71 | 0.92 |
| 120 | ConvNeXt | 0.77 | 0.99 | 0.99 | 0.95 | 1.00 | 0.98 | 0.98 | 0.94 | 0.95 | 0.93 | 0.96 | 0.95 | 0.98 |
| | ViT | 0.82 | 0.99 | 1.00 | 0.53 | 0.99 | 0.99 | 0.98 | 0.95 | 0.80 | 0.80 | 0.89 | 0.82 | 0.98 |
| | EfficientViT | 0.73 | 0.98 | 1.00 | 0.95 | 1.00 | 0.97 | 0.97 | 0.92 | 0.92 | 0.94 | 0.95 | 0.94 | 0.97 |
| | SpecTTTra-$\gamma$ | 0.98 | 0.99 | 1.00 | 0.37 | 1.00 | 0.99 | 0.99 | 0.99 | 0.77 | 0.76 | 0.88 | 0.79 | 0.99 |
| | SpecTTTra-$\beta$ | 0.87 | 0.99 | 1.00 | 0.62 | 0.99 | 0.99 | 0.99 | 0.96 | 0.84 | 0.82 | 0.92 | 0.86 | 0.99 |
| | SpecTTTra-$\alpha$ | 0.78 | 0.99 | 1.00 | 0.96 | 1.00 | 0.99 | 0.99 | 0.98 | 0.89 | 0.87 | **0.97** | 0.96 | 0.99 |

Table 5: Comparison of conventional models and SpecTTTra against human evaluators.

| Partition | | ConvNeXt | ViT | EfficientViT | SpecTTTra-$\gamma$ | SpecTTTra-$\beta$ | SpecTTTra-$\alpha$ | Human |
|---|---|---|---|---|---|---|---|---|
| Algorithm (Sens.) | Suno v2 | 0.65 | 0.80 | 0.65 | 0.54 | 0.64 | 0.72 | 0.69 |
| | Suno v3 | 0.99 | 0.96 | 0.96 | 0.98 | 0.98 | 0.99 | 0.75 |
| | Suno v3.5 | 0.99 | 0.98 | 0.98 | 0.99 | 0.99 | 0.99 | 0.82 |
| | Udio 32 | 0.67 | 0.56 | 0.58 | 0.18 | 0.23 | 0.33 | 0.23 |
| | Udio 130 | 0.99 | 0.87 | 0.98 | 0.99 | 0.99 | 0.99 | 0.55 |
| Fake Type (Sens.) | Half Fake | 0.91 | 0.93 | 0.90 | 0.88 | 0.90 | 0.92 | 0.71 |
| | Mostly Fake | 0.84 | 0.79 | 0.81 | 0.64 | 0.69 | 0.72 | 0.66 |
| | Full Fake | 0.83 | 0.77 | 0.78 | 0.64 | 0.68 | 0.72 | 0.63 |
| Real (Spec.) | | 0.98 | 0.80 | 0.98 | 0.97 | 0.95 | 0.94 | 0.78 |
| Fake (Sens.) | | 0.85 | 0.82 | 0.82 | 0.66 | 0.72 | 0.75 | 0.66 |
| Overall (F1) | | **0.92** | 0.82 | 0.87 | 0.78 | 0.80 | 0.83 | 0.71 |

Table 6: Comparison of SpecTTTra against conventional models on efficiency related metrics.

| Len. (sec) | Model | Speed (A/S) ↑ | FLOPs (G) ↓ | Mem. (GB) ↓ | # Act. (M) ↓ | # Param. (M) ↓ |
|---|---|---|---|---|---|---|
| 5 | ConvNeXt | 137 | 1.5 | 0.4 | 4 | 28 |
| | ViT | 156 | 1.1 | 0.2 | **2** | **17** |
| | EfficientViT | 55 | **0.6** | 0.5 | 5 | 22 |
| | SpecTTTra-$\gamma$ | 154 | 0.7 | **0.1** | **2** | **17** |
| | SpecTTTra-$\beta$ | 152 | 1.1 | 0.2 | **2** | **17** |
| | SpecTTTra-$\alpha$ | 148 | 2.9 | 0.5 | 6 | **17** |
| 120 | ConvNeXt | 39 | 43.1 | 11.7 | 129 | 28 |
| | ViT | 34 | 31.7 | 5.3 | 67 | 17 |
| | EfficientViT | 43 | 15.9 | 14.8 | 138 | **22** |
| | SpecTTTra-$\gamma$ | **97** | **10.1** | **1.6** | **20** | 24 |
| | SpecTTTra-$\beta$ | 80 | 14.0 | 2.3 | 29 | 21 |
| | SpecTTTra-$\alpha$ | 47 | 23.7 | 3.9 | 50 | 19 |

### 4.3.2 HUMAN-AI BENCHMARK

To evaluate Human performance in comparison to AI-based models, we selected a subset of 520 samples from our large test data. This evaluation employed a dynamic scoring system, similar to LMSYS (Chiang et al., 2024), allowing public participation and live leaderboard updates, which will be made publicly available after decision of this paper. Three human participants were involved in this benchmark, with their performance summarized in Table 5. In contrast to the AI benchmark using short (5 sec) or long (120 sec) audio samples, this human benchmark employed 25 sec clips. This choice stems from the observation that short clips hinder human identification due to subtle inaudible artifacts easily detected by AI, while longer clips do not necessarily improve human performance due to how difficult it is to notice long-range temporal dependencies.

As shown in Table 5, AI-based methods consistently outperform human participants across all test partitions. However, both humans and AI models struggle most with Udio algorithms, particularly Udio 32, where human sensitivity dropped to 23%. Conversely, Suno algorithms, especially Suno v3.5, are easier to detect, with a human sensitivity of 82%. This mirrors the findings in the AI benchmark, where models demonstrated higher specificity than sensitivity, indicating greater ease in identifying real songs compared to fake ones. Further analysis revealed distinct patterns within real and fake songs. For instance, Suno algorithms often produced synthetic or mechanical-sounding vocals, while Udio 32 algorithm occasionally created the "*Karaoke effect*." Furthermore, Udio algorithms demonstrated the ability to create songs with multiple voices and higher notes, a feature absent in Suno algorithms. On the other hand, real songs exhibit unique features such as a wide note range, diverse timbre, complex rhythms, clear vocals, and unique sounds like flutes and finger snaps.

### 4.3.3 EFFICIENCY BENCHMARK

To comprehensively evaluate the efficiency of the proposed SpecTTTra model alongside other methods, we measure various metrics across different song lengths using a P100 16GB GPU. The metrics

Table 7: Ablation analysis of temporal and spectral tokens on model performance.

| 5 sec | | | | | 120 sec | | | | |
|---|---|---|---|---|---|---|---|---|---|
| Temp. Clip Size (t) | # Temp. Tok. (T/t) | Spec. Clip Size (f) | # Spec. Tok. (F/f) | F1 | Temp. Clip Size (t) | # Temp. Tok. (T/t) | Spec. Clip Size (f) | # Spec. Tok. (F/f) | F1 |
| 3 | 128 | - | 0 | 0.76 | 3 | 1248 | - | 0 | 0.91 |
| 3 | 128 | 5 | 25 | 0.78 | 3 | 1248 | 5 | 25 | 0.94 |
| 3 | 128 | 3 | 42 | 0.79 | 3 | 1248 | 3 | 42 | 0.96 |
| 3 | 128 | 1 | 128 | **0.80** | 3 | 1248 | 1 | 128 | **0.97** |
| - | 0 | 1 | 128 | 0.75 | - | 0 | 1 | 128 | 0.92 |

considered include Speed (**A/S** → Audio per Second), Floating Point Operations (**FLOPs**), GPU Memory Consumption (**Mem.**) during the forward pass with a batch size of 12, activation count (**# Act.**), and parameter count (**# Param.**). The results are summarized in Table 6. Our analysis reveals that while ViT is the fastest model for 5 sec songs, it becomes the slowest for 120 sec songs (SpecTTTra-$\alpha$ is 38% faster) and exhibits significant memory consumption (SpecTTTra-$\alpha$ uses 26% less memory), rendering it less practical for longer sequences. However, ViT remains the most efficient in terms of parameter count across both short and long songs. On the other hand, ConvNeXt, despite its strong detection performance, becomes very resource-intensive for longer sequences. It consumes a large amount of memory and has the highest FLOPs and parameter count in that category. EfficientViT shows decent performance but with a surprisingly slow speed for short songs, which is over 2x slower than other models. However, in long songs, it shows better speed and lesser FLOPs than ViT and ConvNeXt but has the largest memory requirement and activation count. In contrast, the SpecTTTra model variants excel in their efficiency without compromising competitive performance in longer sequences. For example, in 120 sec songs, the SpecTTTra-$\gamma$ variant emerges as the fastest and most memory-efficient model, being nearly 3x faster and computationally more economical than ViT while showing competitive performance to it. Similarly, the SpecTTTra-$\beta$ variant is more than 2x faster than ViT and uses 2x less memory, all while achieving 3% higher performance. Performance increase culminates in the SpecTTTra-$\alpha$ variant, which outperforms ConvNeXt and EfficinetViT by 1% and 2% respectively, and achieves the highest F1 score of 97%. It achieves this by being 20% and 9% faster while using nearly 67% and 74% less memory, respectively. Therefore, SpecTTTra has the overall best performance while also being the most efficient model in the detection benchmark.

## 4.4 ABLATION STUDY

We conduct an ablation study to highlight the importance of both temporal and spectral tokens, with the findings summarized in Table 7. Additionally, we vary the number of spectral tokens independently of temporal tokens to evaluate their impact on performance. Specifically, we change the clip size $(t, f)$ relative to our best-performing model, SpecTTTra-$\alpha$ ($t = 3, f = 1$), to derive further insights. Notably, while it is possible to classify real and fake songs using only temporal tokens ($F/f = 0$) or only spectral tokens ($T/t = 0$), the combination of both clearly yields the best performance, underscoring their complementary nature. Furthermore, increasing the song duration consistently enhance performance for both spectral and temporal tokens, reinforcing our assertion about the significance of long-context information.

## 5 CONCLUSION

In this paper, we introduced SONICS, a comprehensive dataset for end-to-end synthetic song detection, addressing limitations in existing datasets, such as lack of music diversity, short duration, and most importantly, the absence of end-to-end AI-generated songs. Moreover, we proposed the SpecTTTra model, which efficiently captures long-range temporal relationships in songs, achieving comparable performance to existing popular models while reducing computational costs significantly. Through extensive experiments, we established both AI-based and human benchmarks, demonstrating the dataset's effectiveness in advancing synthetic song detection research. This work paves the way for future research to more effectively distinguish AI-generated music, thereby aiding in the preservation of human musical artistry.

## 6 ETHICS STATEMENT

The dataset incorporates copyrighted song data from YouTube. To comply with legal standards, we provide only YouTube links to the original songs and will publicly release only the AI-generated songs. Given that the generative models used to generate these fake songs may have been trained on copyrighted songs, there could be potential concerns regarding the copyright status of our dataset. However, even if the generative models were trained using copyrighted data, our dataset falls under the fair use policy U.S. Code (2023) according to research criteria. The same is true for additional metadata that was used to generate the dataset, including lyrics, style, etc. Thus, the use of these models and relevant metadata for generating our dataset is justified. Furthermore, the practice of using generative models (which likely have been trained on copyrighted data) to create large-scale datasets has been documented in the literature and published in peer-reviewed, widely accepted conferences. Notable examples include LLaVA-Instruct-158K (from GPT-4) Liu et al. (2024), Gpt4tools (from ChatGPT) Yang et al. (2024), Camel (from GPT-3.5 Turbo) Li et al. (2023), the Baize Dataset (from ChatGPT) Xu et al. (2023), and the JourneyBench Dataset (which uses GPT-4V and GPT-4O) Wang et al. (2024), among others.

Additionally, there might be concerns that these generated models regurgitate copyrighted music. To appease these concerns and verify whether the generated songs are identical to existing real songs, we compared all the generated songs with real songs in a pairwise manner using the cosine similarity metric with EfficientNetB0 embeddings as representations. Among the top 50 songs with the highest similarity to real songs, we manually inspected each one and found no exact matches. All these songs exhibited variation in elements such as music style, vocals, instrumentation, or other features, including the "half fake" subset, where only the lyrics were shared. Given this variation, our work also falls under fair use policies U.S. Code (2023). Finally, as these fake songs are generated through paid subscriptions that allow for the use and sharing of content, our dataset will be made publicly available under a `CC BY-NC 4.0` license.

We also acknowledge issues related to bias and fairness. The dataset is currently limited to English-language songs, which affects its global applicability. Future work will address this by expanding the dataset to include more languages. Notably, a gender bias is evident, with male singers dominating the song styles, a trend that may stem from either the real songs or the Gemini 1.5 Flash model that was used to extract song styles (Half Fake and Mostly Fake songs), or GPT-4 that was used to generate song styles (Full Fake Songs). Addressing this gender bias is beyond the scope of our study, and we leave it to the community to tackle in future research.

## 7 REPRODUCIBILITY STATEMENT

To ensure reproducibility, we have made extensive efforts to document and share all necessary details. First, we provide the complete dataset generation process, including the end-to-end pipelines. The Appendix offers additional information, such as dataset statistics, to help better understand the data. Second, the pseudo-code for the Spectro-Temporal Tokenizer of the SpecTTTra model is presented in the Appendix. All hyperparameters, training setups, and augmentation methods are detailed in the "Implementation Details" section of both the main paper and the Appendix. Third, we include all assumptions and configurations for the benchmarks, which are available in both the main paper and the Appendix. Finally, the source code is provided in the supplementary materials, which contains detailed configurations for training, model parameters, and profiling.

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

# Appendix

# Contents

## A  DATASET CHARACTERISTICS

**Long Form Correlations:** An example of the long context correlation can be observed on Fig. 5, where a real song would stay consistent throughout the repetition of the phrases, refrains, rythms, etc. In comparison, synthetic generation methods can have difficulty generating consistent refrains, due to the long form context that might be required to model this information. Therefore, a synthetic song detection model that can capture the nuances of long context correlations present in songs would be able to better differentiate real and synthetic songs utilizing this property.

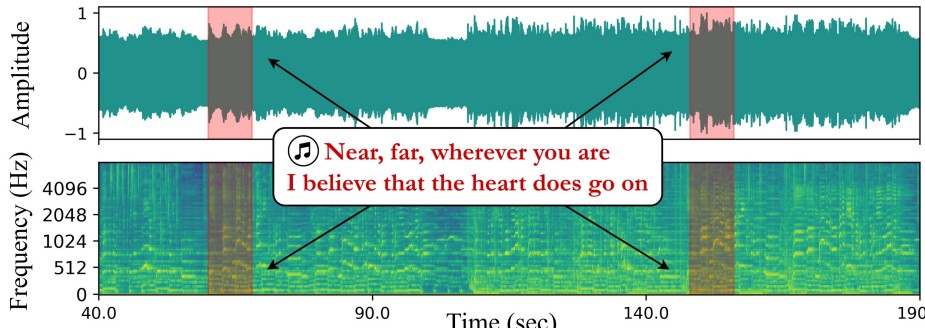

Figure 5: Long Context Correlation. The red highlighted regions in the spectrogram and raw audio indicate repetition of the same verse (`"Near, far, wherever you are, I believe that the heart does go on"`), rhythms, and music. Such consistency is a characteristic of real songs and can be challenging for synthetic generation methods to replicate.

**Song Duration:** The distribution of the song duration across training and test sets is shown on Fig. 6. From the figure, it can be observed that aside from a few minor outliers, the distribution of fake and real song duration across both training and test sets are similar.

**Genre Distribution:** Fig. 7 reveals that, despite slight imbalances, both real and fake songs are represented across all genres. This observation also indicates that the detectors are not overfitting to specific song genres. In other words, the models are not relying on genre information to determine whether a song is real or fake. Because if the detectors were attempting to perform genre classification as a shortcut for detecting fake songs, their performance would have deteriorated. However, the strong performance of all methods demonstrates that our models are effectively learning features beyond genre classification. This finding provides confidence that the detection capability is robust and not dependent on genre-specific cues.

**Embedding Space:** The t-SNE plots presented in Fig. 8 illustrate the data distributions within the embedding space, generated using an EfficientNetB0 (Tan & Le, 2019) encoder with Mel Spectrogram audio inputs.

- **Subfigure (a)** shows a significant overlap between the training (Green) and testing (Orange) datasets. However, distinct clusters along the edges, predominantly composed of training data, suggest that the training set is more diverse than the test set. This diversity is observed despite the test set containing previously unseen algorithms and speaker scenarios.

- **Subfigure (b)** indicates a substantial overlap between real (Red) and fake (Blue) songs, with a few regions exclusively occupied by fake data or outliers. This highlights the inherent challenge of synthetic song detection within this dataset.

- **Subfigure (c)** shows that the embedding space is primarily occupied by Suno v3.5 (Green) and Udio 133 (Orange), which are seen algorithms. Despite Suno v2 (Blue), Suno v3 (Red), and Udio 32 (Violet) being unseen algorithms, their embeddings fall within regions covered by seen algorithms, demonstrating underlying similarities. A noteworthy observation is the considerable overlap between Suno v3 (Red) and Suno v3.5 (Green), which likely contributes to the similar detection performance in these algorithms, as indicated by Table 4.

- **Subfigure (d)** depicts the distribution of fake types. Although Half Fake (Orange) songs form discernible clusters, Full Fake (Red) songs exhibit a more scattered distribution, in-

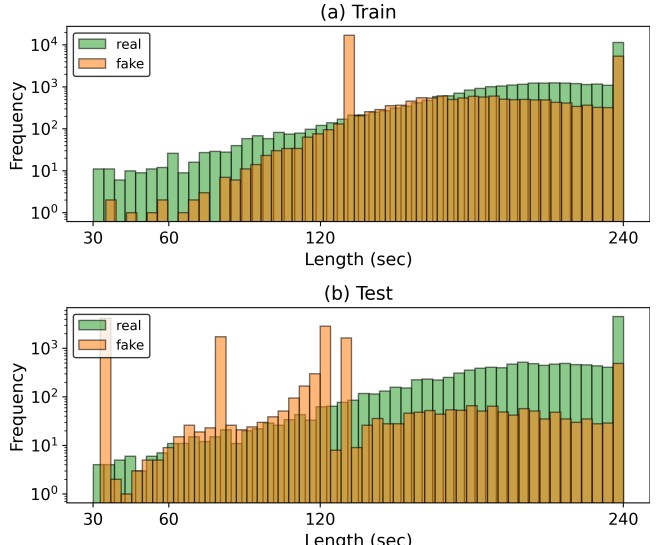

Figure 6: Duration distribution between Training and Testing sets.

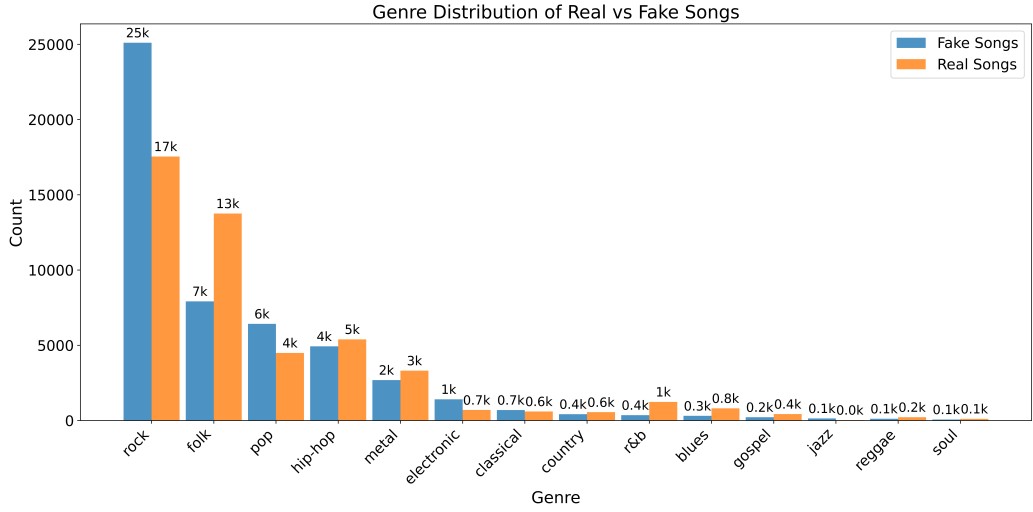

Figure 7: Genre Distribution between Real and Fake Songs.

dicating a lack of common features. Conversely, Mostly Fake (Green) songs are spread broadly across the embedding space, reflecting their diverse characteristics.

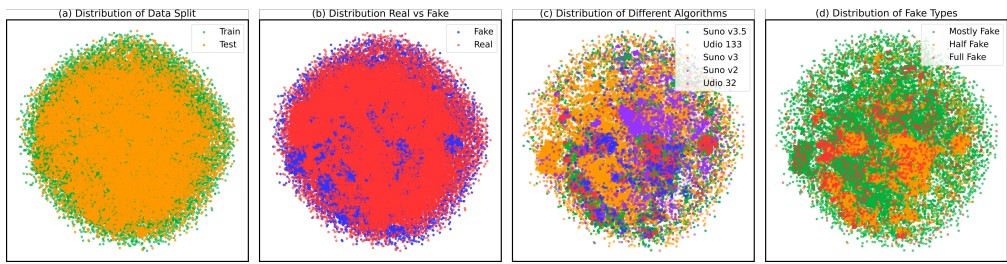

Figure 8: Data distributions in the embedding space. This plot illustrates the clustering of different data categories such as target, algorithms, fake types, and data split.

**Song Style:** In order to imitate the song style observed in real songs, we had LLMs perform a stylistic analysis of the real song lyrics (prompt shown in Table 12). The wide variety of styles extracted from real songs are utilized to generate the synthetic songs. It is important to have the distribution of synthetic song style as close to the real song styles in the dataset, since it would be a better indicator of real life cases. A wide diversity of synthetic song style in training data is also required so that the models trained on this dataset do not rely on the distinct characteristics of any particular song style as a feature to detect synthetic songs. Additionally, it is also important to evaluate performance of the trained model on a testing set across a wide variety of song style as well, so that the generalization capability of the detection model across multiple styles is evaluated. A word cloud representation of the song styles that have been extracted from real songs and utilized to generate songs is show on Fig. 9. The figure indicates that there are diverse song styles across both training and testing sets.

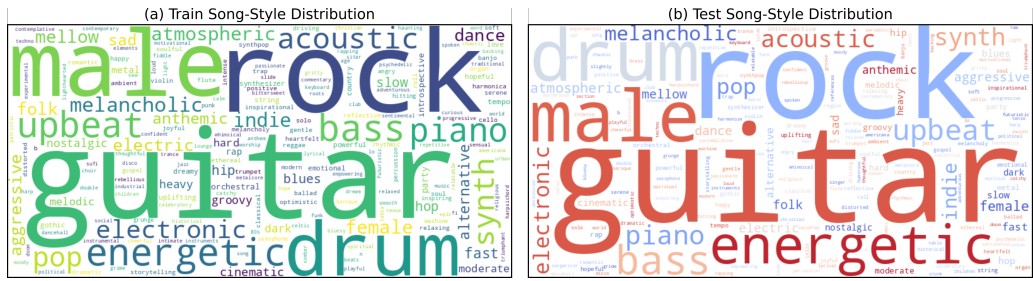

Figure 9: Data distributions of song-style between train and test split in the word cloud.

## B    DATASET QUALITY EVALUATION

Given the large scale of the datasets, exhaustive manual verification is not feasible. However, we have rigorously evaluated our pipeline and found that any potential errors have minimal impact on the dataset's overall quality. For instance, to assess PyAnnotate's performance in detecting non-vocal segments, we manually analyzed 50 random samples from our dataset. The results showed that the tool is highly accurate in distinguishing between vocals and non-vocals. While some errors may exist, their impact is negligible. Specifically, human evaluators reported no instances of songs containing only background music without vocals (i.e., no false negatives). Furthermore, the total number of samples classified as non-vocal (false positives) by PyAnnotate is only 3% of the dataset. Even if we conservatively assume that all of these are false positives—though this is not the case based on our evaluation—the overall impact on the dataset remains minimal. Additionally, to assess the accuracy of the extracted song styles, we manually analyzed 50 randomly selected real songs and compared their AI-generated styles (genre, instruments, mood, and vocal type) with their actual styles, as retrieved from the internet. Our analysis showed that the generated styles were highly accurate for genre, instruments, and mood. However, occasional inaccuracies were observed in the vocal type classification, which will not impact the primary goal of our work—fake song detection.

## C    IMPLEMENTATION DETAILS

**Dataset Cost Analysis:** Generating the SONICS dataset costs $1,055, allocated across GPT-4o ($405), Gemini 1.5 Flash ($80), Suno ($390), and Udio ($180).

**SingFake Training:** To demonstrate the limitations of SVDD (Singing Voice Deepfake Detection) models in detecting end-to-end fake songs, we first trained the models on the SingFake dataset, which consists of 597 training songs and 480 testing/validation songs. We then evaluated the performance of these models on our SONICS-test dataset, with the results summarized in Table 2. It is important to note that due to some inactive links in the SingFake dataset, we were able to download only 1077 songs, as opposed to the 1305 songs originally reported in the paper. For training, we maintained the same model configuration as in other experiments, with one exception: we resized the $128 \times 128$ (5 sec) Mel spectrograms to $224 \times 224$ to leverage the pretrained ImageNet weights, compensating for the smaller size of the SingFake dataset.

**Augmentation:** To enhance the robustness of our models, we apply MixUp (Zhang, 2017) augmentation with $\alpha = 2.5$ and a 50% probability. Additionally, we utilize SpecAugment (Park et al., 2019), applying two time masks of size 8 and one frequency mask of size 8, each with a 50% probability.

**Benchmark:** For the efficiency benchmark, we utilized a single P100 GPU for all experiments. To measure the inference time of each model, we performed 5 warm-up runs followed by 100 test runs with a batch size of 1 to record the processing time. The results were averaged and then inverted to determine the inference speed of each model. To compute GPU memory consumption, we used a batch size of 14 across all models, measuring the peak memory usage during a single forward pass. For calculating FLOPs, we employed the `fvcore` (FAIR, 2023) library.

**Model:** For the proposed SpecTTTra model, we use the *vit_small_patch16* configuration from timm (Wightman, 2019) library with an embedding dimension of 384, 6 attention heads, 12 Transformer layers, and an MLP ratio of 2.67.

To clarify the core components of the SpecTTTra model, we provide PyTorch-like code for the Spectro-Temporal Tokenizer ($st_tokenizer$) below. Note that while the code is presented in a functional format for clarity, the actual implementation follows an object-oriented approach.

```python
import torch
import torch.nn as nn

def st_tokenizer(x, t_clip, f_clip, embed_dim):
    B, F, T = x.size()

    # Temporal tokens
    t_tokens = tokenizer(x, F, embed_dim, t_clip, T // t_clip)

    # Spectral tokens
    f_tokens = tokenizer(
        x.transpose(1, 2),
        T,
        embed_dim,
        f_clip,
        F // f_clip,
    )

    # Spectro-Temporal tokens
    st_tokens = torch.cat((t_tokens, f_tokens), dim=1)
    return st_tokens

def tokenizer(x, input_dim, token_dim, clip_size, n_clips):
    # Slicing and Tokenization
    conv1d = nn.Conv1d(
        in_channels=input_dim,
        out_channels=token_dim,
        kernel_size=clip_size,
        stride=clip_size,
        bias=False,
    )
    x = conv1d(x).gelu().transpose(1, 2)

    # Positional Embedding
    pos_embeds = nn.Parameter(torch.randn(1, n_clips, token_dim) * 0.02)
    x = x + pos_embeds

    # Layer Normalization
    x = nn.LayerNorm(token_dim, eps=1e-6)(x)
    return x
```

In this code, the $tokenizer$ function represents the spectral or temporal tokenizer used to embed spectral or temporal clips (patches) into tokens. Here, $f\_clip$ and $t\_clip$ denote the sizes of the spectral and temporal clips, respectively, while $embed\_dim$ signifies the feature dimension of each token. The dimensions $T$ and $F$ correspond to the temporal and spectral dimensions of the input spectrogram.

## D  BENCHMARK

### D.1  GENERALIZATION TEST

Although generalization is not the primary focus of our work, it remains crucial for evaluating the broader applicability of our dataset and methods. To assess generalization, we evaluated models trained on the SONICS dataset using out-of-distribution (OOD) songs from two external generators: SkyMusic and SeedMusic. Due to the lack of an API for SkyMusic and a platform for SeedMusic, we manually collected 394 and 36 songs, respectively, from their publicly available demo websites. While the sample size is limited, these tests offer preliminary insights into generalization performance. The results, summarized in Table 8, reveal the following key observations:

1. **Performance Decline:** A consistent decline in performance is evident across all models, as shown in Table 8. This trend aligns with findings in media forensics Ojha et al. (2023); Epstein et al. (2023); Yan et al. (2024), underscoring the challenges of generalization. For instance, ConvNeXt, which performed exceptionally well on the SONICS dataset, exhibited significant performance drops, ranking the lowest across both sources and durations.

2. **SpecTTTra's Robustness:** SpecTTTra-$\gamma$ achieved the highest F1 scores (80% on Seed-Music and 60% on SkyMusic), outperforming all other models. However, larger variants, such as SpecTTTra-$\alpha$, showed susceptibility to overfitting, particularly with long-duration songs.

3. **Transformer Advantage:** Models incorporating transformers (e.g., SpecTTTra, ViT, EfficientViT) demonstrated greater resilience to OOD songs compared to CNN-based models, reaffirming their suitability for addressing generalization challenges.

The observed performance declines on OOD songs are consistent with prior findings Ojha et al. (2023); Epstein et al. (2023); Yan et al. (2024), emphasizing the necessity of advanced generalization strategies. Despite the absence of such strategies, some models achieved promising OOD results, highlighting the potential of the SONICS dataset for advancing song forensics. Notably, for validation purposes, we also evaluated all models with randomly initialized weights. None of these models exceeded an F1 score of 0.51, further reinforcing the significance of learned representations.

### D.2  HUMAN-AI BENCHMARK

To assess human performance in synthetic song detection, we developed a Huggingface space called "Song Arena" as illustrated in Fig. 10. In this space, users can evaluate whether a randomly selected song from a subset of the proposed dataset (comprising 520 samples) is synthetic or not. The space also features a leaderboard (shown in Fig. 11) that records human detection performance for songs generated by different algorithms and generation methods. The evaluation metrics used to assess the detectability of synthetic songs include the F1 score, Sensitivity (True Positive Rate), and Specificity (True Negative Rate). These metrics provide a comprehensive measure of the difficulty humans face in detecting synthetic songs generated by various algorithms.

## E  RESULT ANALYSIS

Our analysis of the SpecTTTra-$\alpha$ model's results reveals audible and perceptible artifacts in both successful and failed cases of real and fake songs. In True Negative cases, we find distinct patterns in correctly classified real songs. These include characteristics such as unpredictability, dynamic variation, and unexpected changes that is often absent in fake songs. Examples include non-standard pitch variations, intricate rhythmic complexity, and expressive techniques like melismatic phrasing, sudden tempo changes, or improvisational segments, all of which showcase the nuanced artistry of human performance. Conversely, in True Positive cases, we detect specific audible artifacts in

Table 8: Comparison of SpecTTTra and Existing Methods on Unseen Generators (SkyMusic and SeedMusic). AASIST and its variants face out-of-memory error while training for 120s audio samples resulting in missing performance scores (-).

| Len. (sec) | Model | Seed Music | Sky Music |
|---|---|---|---|
| 5 | AASIST | 0.65 | 0.56 |
| | ResNet + Spec. | 0.52 | 0.48 |
| | ResNet + LFCC | 0.54 | 0.51 |
| | Wav2Vec2 + AASIST | 0.69 | 0.53 |
| | ConvNeXt | 0.28 | 0.21 |
| | ViT | 0.68 | 0.57 |
| | EfficientViT | 0.67 | 0.54 |
| | SpecTTTra-$\gamma$ | 0.32 | 0.33 |
| | SpecTTTra-$\beta$ | 0.32 | 0.41 |
| | SpecTTTra-$\alpha$ | 0.47 | 0.57 |
| 120 | AASIST | - | - |
| | ResNet + Spec. | 0.53 | 0.35 |
| | ResNet + LFCC | 0.56 | 0.40 |
| | Wav2Vec2 + AASIST | - | - |
| | ConvNeXt | 0.36 | 0.19 |
| | ViT | 0.64 | 0.53 |
| | EfficientViT | 0.71 | 0.55 |
| | SpecTTTra-$\gamma$ | **__0.80__** | **__0.60__** |
| | SpecTTTra-$\beta$ | 0.53 | 0.28 |
| | SpecTTTra-$\alpha$ | 0.56 | 0.25 |

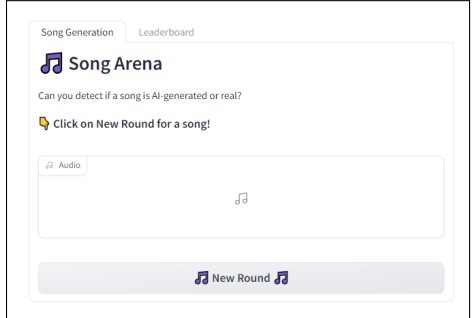
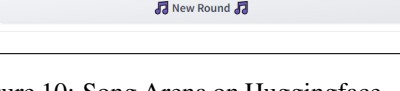

Figure 10: Song Arena on Huggingface.

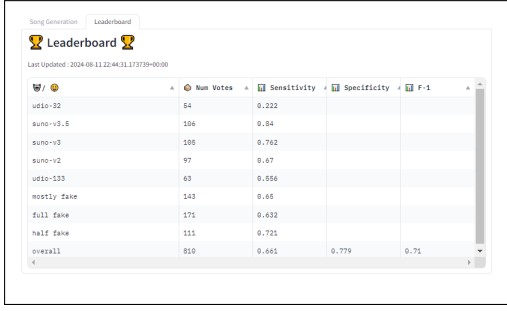

Figure 11: Song Arena Leaderboard on Huggingface.

correctly classified fake songs. These artifacts include mechanical or robotic vocal qualities, unclear vocal articulation, predictable rhythmic structures, and limited pitch variability. Such fake songs also lack the emotive expressiveness and complexity we consistently find in real music, making them notably distinct. In False Negative cases, we observe that fake songs not detected as such lack the typical artifacts seen in true positive fake songs. Instead, these cases often incorporate features that mimic the unpredictability and nuanced variation of real songs. For instance, some include spoken interludes or conversational segments before the singing starts, creating a deceptive resemblance to genuine music. In False Positive cases, we find that real songs misclassified as fake share characteristics with AI-generated music. These include unclear vocals, less rhythmic complexity, and a noticeable absence of the unexpected changes that typically distinguishes human-created performances. Finally, we encounter instances where we are unable to detect any audible artifacts. This suggests the presence of subtle, imperceptible, or inaudible artifacts (Barrington et al., 2023) akin to invisible artifacts (Chhabra et al., 2023) in synthetic images.

## F    COMPARISON WITH RELATED WORKS

To comprehensively evaluate our proposed method, we compared our SpecTTTra models with various related works in fake song detection and provided our findings in Table 9. Specifically, we benchmarked our method against all the approaches mentioned in SingFake Zang et al. (2024b), including AASIST Jung et al. (2022), Wav2Vec Baevski et al. (2020), and ResNet He et al. (2016) variants. Our analysis demonstrates that the SpecTTTra models outperform the compared methods in terms of effectiveness while maintaining efficiency. For example, although the AASIST models deliver promising results for short songs, they are computationally intensive, as reflected by their high time and memory consumption. Additionally, AASIST models fail to process long songs of 120 seconds, resulting in out-of-memory (OOM) errors due to its extreme computational complexity. On the other hand, while ResNet variants exhibit high efficiency, they lack the capacity to detect fake songs as effectively as SpecTTTra, particularly with long-duration songs. In contrast, the SpecTTTra models strike a balance between efficiency and effectiveness. As shown in the Table 9, SpecTTTra-$\alpha$ achieves the highest F1 score of 0.97 for long-duration songs while maintaining manageable computational requirements. These results establish the superiority of our method in both detecting fake songs and handling longer audio inputs compared to other existing methods.

Table 9: Performance and efficiency comparison with existing works in fake song detection. AASIST and its variants face out-of-memory error while training for 120s audio samples resulting in missing performance scores (-). Their FLOPs and activation count were also unable to be determined by the fvcore library.

| Len. (sec) | Model | Speed (A/S) ↑ | FLOPs (G) ↓ | Mem. (GB) ↓ | # Act. (M) ↓ | # Param. (M) ↓ | F1 ↑ |
|---|---|---|---|---|---|---|---|
| 5 | AASIST | 55 | 12 | 6 | 96 | 0.3 | 0.91 |
| | ResNet + Spec. | 354 | 0.6 | 0.1 | 0.8 | 11 | 0.86 |
| | ResNet + LFCC | 331 | 0.6 | 0.1 | 0.8 | 11 | 0.88 |
| | Wav2Vec2 + AASIST | 42 | * | 1.3 | * | 95 | 0.90 |
| | SpecTTTra-$\gamma$ | 154 | 0.7 | 0.1 | 2 | 17 | 0.76 |
| | SpecTTTra-$\beta$ | 152 | 1.1 | 0.2 | 2 | 17 | 0.78 |
| | SpecTTTra-$\alpha$ | 148 | 2.9 | 0.5 | 6 | 17 | 0.80 |
| 120 | AASIST | 2 | 295 | OOM | 2393 | 0.3 | - |
| | ResNet + Spec. | 146 | 17.2 | 2.3 | 24 | 11 | 0.90 |
| | ResNet + LFCC | 144 | 17.2 | 2.3 | 25 | 11 | 0.91 |
| | Wav2Vec2 + AASIST | 2 | * | OOM | * | 95 | - |
| | SpecTTTra-$\gamma$ | 97 | 10.1 | 1.6 | 20 | 24 | 0.88 |
| | SpecTTTra-$\beta$ | 80 | 14.0 | 2.3 | 29 | 21 | 0.92 |
| | SpecTTTra-$\alpha$ | 47 | 23.7 | 3.9 | 50 | 19 | 0.97 |

## G    LIMITATIONS AND FUTURE WORK

The proposed SONICS dataset contains real songs dynamically queried from YouTube using their titles and artist names, which can sometimes result in incorrect audio retrieval. A manual analysis of 600 random samples suggests that this issue affects approximately 0.5% of the dataset. To address this minor noise, we utilized label smoothing (Szegedy et al., 2016) during training. Another limitation is that the fake songs generated by the Udio platform cannot include lyrics from real songs, limiting the comprehensive evaluation in the Half Fake songs category to only those generated by the Suno platform. Our current benchmarks are based solely on Mel Spectrogram inputs; hence, we aim to incorporate raw audio and explore other feature extraction methods, such as LFCC and MFCC, to enhance the robustness of our evaluations. Due to resource constraints, we trained and compared only smaller versions of all models. In the future, we plan to compare larger versions of all models. Furthermore, we trained all models from scratch to ensure a fair comparison, because the proposed SpecTTTra model is designed specifically for audio, while other models like ConvNeXt and ViT only have pretrained weights available for images (ImageNet). In the future, we plan to pretrain all models on an large-scale audio dataset from scratch before training them on our proposed dataset for maximizing the performance.

## H  PROMPT ENGINEERING

**Selection of LLMs**: For lyrics and song-style generation, as well as lyrics feature extraction, we evaluated several proprietary LLMs, including GPT-4o, Claude 3, and Gemini 1.5, along with open-source models like LLama 3, Gemma 2, and Mistral Large. Among these models, GPT-4o demonstrated superior performance, particularly in maintaining rhythm and coherence and accurately following the content of prompts. Based on these qualities, we selected GPT-4o for both lyrics and song-style generation tasks.

For song-style analysis, only the proprietary models Gemini 1.5 Pro and Gemini 1.5 Flash, as well as the open-source model Qwen-Audio (Chu et al., 2023), are equipped to process audio inputs. Our evaluations indicated that both Gemini 1.5 Pro and Gemini 1.5 Flash models deliver similarly accurate performance. In contrast, Qwen-Audio frequently struggled to follow instructions correctly. Considering that the Gemini 1.5 Flash model is ten times more cost-effective than the Gemini 1.5 Pro, we selected the Gemini 1.5 Flash model for song-style analysis.

**Half Fake:** These songs are generated using lyrics and song style extracted from real songs. To extract the song style, the prompt template mentioned in Table 12 is used. This template extracts song style information such as vocal type, musical instruments, mood, etc.

**Mostly Fake:** These songs are generated similarly to Half Fake songs, except that the lyrics are AI-generated. The lyrics are created using an LLM (Large Language Model) with the prompt template shown in Table 11, where lyrics features are used as input. These lyrics features are also extracted from real song lyrics using the prompt template shown in Table 10. The use of lyrics features instead of direct lyrics prevents the LLM from copying the original content, encouraging it to generate lyrics with a similar distribution rather than duplicating them.

**Full Fake:** These songs are generated using AI-generated lyrics and song style, created using the prompt template provided in Table 13. In this process, the genre, topic, and mood were selected randomly from the lists provided below:

- **List of Genres:** `alternative, baroque, blues, bollywood, c-pop, celtic, christian rock, classical, country, crunk, dance, dancehall, disco, doom metal, electronic, folk, funk, fusion, gospel, gothic, grime, grunge, hard rock, heavy metal, hip hop, indie rock, j-pop, jazz, k-pop, lo-fi, lounge, metal, metalcore, new age, opera, orchestral, pop, pop rock, progressive metal, progressive rock, punk, r&b, rap, reggae, salsa, smooth jazz, soul, sufi, world music.`

- **List of Moods:** `adventurous, ambivalent, amused, angry, anxious, apathetic, bittersweet, blissful, calm, carefree, cautious, chaotic, confident, confused, curious, desperate, determined, disenchanted, distracted, drained, dreamy, empathetic, enchanted, energetic, exhilarated, focused, forgiving, frustrated, gloomy, grateful, hateful, humble, inspired, introspective, jealous, joyful, liberated, lonely, loving, melancholic, mischievous, motivated, mournful, mysterious, nostalgic, optimistic, passionate, pensive, pessimistic, playful, powerless, proud, rebellious, regretful, reluctant, restless, romantic, sarcastic, satisfied, shocked, skeptical, submissive, sympathetic, tense, timid, trapped, uninspired, vengeful, vulnerable, whimsical, yearning, zealous.`

- **List of Broad Topics:** `alien invasion, ancient civilizations, augmented reality, betrayal, childhood memories, climate change, cyber crime, dimensional portals, dreams and aspirations, dystopian future, empowerment, endangered species, extraterrestrial contact, family, fashion, financial struggles, first kiss, forgiveness, friendship, futuristic cities, generation gap, grief and loss, heartbreak, social media anxiety, interstellar travel,`

loneliness in a crowd, long-distance relationships, love,
love at first sight, lunar colonization, nanotechnology,
nature's beauty, nostalgia, ocean exploration, overcoming
adversity, pandemic experiences, parallel universes,
political revolution, politics, quantum physics,
reincarnation, never give up, road trip adventures,
robotic emotions, save the planet, sibling rivalry, social
influencers, social justice, space exploration, space
tourism, survival in the wild, technology addiction, time
capsules, time paradoxes, time travel, unconditional love,
work-life balance.

- **List of Specific Topics:** elon musk vs yann lecun (AI), yann lecun vs
geoffrey hinton (AI), convolution vs transformer (AI), gan vs
diffusion models (AI), tensorflow vs pytorch (AI), pytorch vs
jax (AI), twilight zone (TV), star trek (TV), game of thrones
(TV), breaking bad (TV), stranger things (TV), big bang theory
(TV), friends (TV), simpsons (TV), house of cards (TV), how I
met your mother (TV),  the office - US (TV),  sherlock (TV),
avatar - the last airbender (TV),  pokemon (anime),  dragon
ball z (anime), naruto (anime), one piece (anime), attack on
titan (anime), my hero academia (anime), death note (anime),
jujustu kaisen (anime),  fullmetal alchemist (anime),  demon
slayer (anime),  neanderthals (anthropology),  pyramids of
giza (archaeology),  machu picchu (archaeology),  stonehenge
(archaeology),  egyptian mummies (archaeology),  giza sphinx
(archaeology),  notre dame cathedral (architecture),  london
bridge (architecture),  big ben (architecture),  eiffel tower
(architecture),  versailles (architecture),  arc de triomphe
(architecture),  mona lisa (art),   van gogh's starry night
(art), sistine chapel (art), solar eclipses (astronomy), black
holes (astronomy),  big bang theory (astronomy),  supernovas
(astronomy),  dark matter (astronomy),  andromeda galaxy
(astronomy),  elon musk vs bezos (business),  taylor swift
(celebrity),  tom cruise (celebrity),  brad pitt (celebrity),
angelina jolie (celebrity),   jennifer aniston (celebrity),
leonardo dicaprio (celebrity),  meryl streep (celebrity),
robert de niro (celebrity),   michael jackson (celebrity),
matt damon vs jimmy kimmel (celebrity),   jimmy kimmel vs
jimmy fallon (celebrity),  marvel vs dc (comics),  batman
vs superman (comics),  justice league vs avengers (comics),
thor vs hulk (comics),  iron man vs captain america (comics),
batman vs joker (comics), john constantine (comics), marvel
universe (comics),  dc comics (comics),  big bang theory
(cosmology),  russian ballet (culture),  bollywood (culture),
hollywood (culture),    alphago - the movie (documentary),
the great hack (documentary),  walt disney (entertainment),
rose bowl parade (event),  coachella (event),  wimbledon
(event),  kentucky derby (event),  rio olympics (event),
tokyo olympics (event), beijing olympics (event), columbus
(exploration),  marco polo (exploration),  nike vs adidas
(fashion),  world war III (fiction),  pepsi vs coke (food),
messi vs ronaldo (football),  pele vs maradona (football),
brazil vs argentina (football), monopoly (game), chess (game),
go (game),  dungeons and dragons (game),  minecraft (game),
fortnite (game), call of duty (game), mario (game), pac-man
(game),  sonic the hedgehog (game),  fifa (game),  cyberpunk
2077 (game), grand theft auto (game),  the last of us (game),
assassin's creed (game),  resident evil (game),  halo (game),

the witcher 3 (game), god of war (game), alphago vs lee sedol (games), deep blue vs kasparov (games), playstation vs xbox (gaming), mount everest (geography), grand canyon (geography), dead sea (geography), ring of fire (geography), venetian canals (geography), rocky mountains (geography), volcanic eruptions (geology), ice age (geology), mariana trench (geology), great depression (history), moon landing (history), titanic (history), viking explorers (history), stone age (history), aztec empire (history), mayan calendar (history), vikings (history), cold war (history), space race (history), moon landing (history), fall of the berlin wall (history), industrial revolution (history), edison's light bulb (invention), the wright brothers (invention), tesla vs edison (inventors), taj mahal (landmark), great wall of china (landmark), central park (landmark), mount fuji (landmark), fifa world cup (sports), empire state building (landmark), statue of liberty (landmark), hollywood sign (landmark), golden gate bridge (landmark), niagara falls (landmark), times square (landmark), king arthur (legend), robin hood (legend), loch ness monster (legend), yeti (legend), camelot (legend), shakespearean sonnets (literature), harry potter (literature), the hobbit (literature), lord of the rings (literature), edgar allan poe (literature), charles dickens (literature), dracula (literature), frankenstein (literature), the great gatsby (literature), pride and prejudice (literature), alice in wonderland (literature), romeo and juliet (literature), moby dick (literature), war and peace (literature), little women (literature), treasure island (literature), oliver twist (literature), peter pan (literature), narnia (literature), aesop's fables (literature), sherlock holmes (literature), the da vinci code (literature), CNN vs Fox News (media), godzilla (movie), inception (movie), matrix (movie), interstellar (movie), john wick (movie), jason bourne (movie), james bond (movie), spiderman (movie), dark knight (movie), avengers (movie), star wars (movie), indiana jones (movie), back to the future (movie), jurassic park (movie), avatar (movie), wizard of oz (movie), star wars vs star trek (movies), star wars (movies), indiana jones (movies), titanic (movies), fight club (movies), the dark knight (movies), the green mile (movies), gladiator (movies), the departed (movies), the lion king (movies), aladdin (movies), beauty and the beast (movies), little mermaid (movies), frozen (movies), tangled (movies), mulan (movies), sleeping beauty (movies), cinderella (movies), snow white (movies), the social network (movies), the beatles (music), bob dylan (music), bermuda triangle (mystery), crop circles (mystery), atlantis (myth), hercules (myth), perseus (myth), pandora's box (myth), trojan war (myth), achilles (myth), hades (myth), olympus (myth), zeus (myth), hera (myth), apollo (myth), artemis (myth), athena (myth), poseidon (myth), phoenix (mythical creature), medusa (mythical creature), greek mythology (mythology), yellowstone (national park), yosemite (national park), grand tetons (national park), amazon rainforest (nature), sahara desert (nature), great barrier reef (nature), victoria falls (nature), niagara falls (nature), northern lights (nature), southern lights (nature), bioluminescent bays (nature), blue holes (nature), dinosaurs (paleontology), democrats vs republicans (politics), scientist vs engineer (profession), python vs c++ (programming), java vs python (programming), javascript vs

**Instructions:**
Analyze the provided song lyrics and extract the following elements:
1. Subject Matter: Write what the song is about by providing a summary of the story, narrative, central topic, or events discussed in the song lyrics.
2. Theme: Identify the main theme or message (emotion, life experience, social commentary, or philosophical concept).
3. Target Audience: Define the intended audience (age group, cultural background, or specific interests).
4. Narrative: The story and point of view (e.g., first person or second person).
5. Character Analysis: Main characters (e.g., Protagonist, Antagonist) and their traits.
6. Song Structure: Outline the structure (number of verses, choruses, bridges, intros, outros) and note any unique elements or deviations.
7. Mood: Describe the overall mood (upbeat, melancholic, introspective, etc.).
8. Reference: Identify any cultural, social, time, place, or contextual references.

**How to respond:**
You should provide your answer below after the "Answer" section. You are not allowed to use any text formatting (bold, italic, etc.) and narrative ('Here is the answer', 'Below is the response', etc.) in your answer. Only answer using the following format:
* subject_matter: "....."
* theme: "...."
* target_audience: "...."
* narrative: "...."
* character_analysis: "...."
* song_structure: "...."
* mood: "...."
* reference: "...."

**Lyrics:**
{lyrics}

**Answer:**

Table 10: Prompt template for extracting lyrics features (e.g subject matter, theme, mood) from real songs. Here, {lyrics} indicates placeholder for input lyrics.

```
java (programming),  paper book vs e-book (reading),  physics
(science),  chemistry (science),  biology (science),  astronomy
(science), mathematics (science), albert einstein (scientist),
isaac newton (scientist),  charles darwin (scientist),  marie
curie (scientist),    summer vs winter (season),    twitter vs
facebook (social media), olympics (sports), world cup (sports),
super bowl (sports),    tour de france (sports),    wimbledon
(sports), nba finals (sports), nfl playoffs (sports), elon musk
vs mark zuckerberg (tech),   ai vs human intelligence (tech),
nvidia vs amd (tech),   intel vs amd (tech),   nvidia vs intel
(tech), google vs openai (tech), iphone vs android (tech), mac
vs pc (tech), ai revolution (tech), rubik's cube (toy), barbie
(toy),  lego (toys),  europe vs asia (travel),  usa vs canada
(travel),   australia vs new zealand (travel),   usa vs europe
(travel),  switzerland vs sweden (travel), beach vs mountains
(vacation), city vs countryside (vacation).
```

**Task:**
You are a talented songwriter tasked with creating a song based on the following lyrics features. The song must include all the features described below in the "Lyrics Features" section. The song should not be long; rather, it should be medium-length.

**Lyrics Features:**
{lyrics_feature}

**Instructions:**
You should write the song with metatags following the song structure from "Lyrics Features". In some very rare cases, you can also scarcely include ad libs, or non-lexical vocables.
* You can add metatags to your lyrics on top of a section in `[square brackets]` that will create certain styles. Some examples of metatags are `[Verse]`, `[Chorus]`, `[Bridge]`, `[Solo]`, `[Outro]`, `[Pre-Chorus]`, `[Bridge]`, `[Hook]`, `[Opening]`, `[Intro]`, `[Instrument]`, `[Build]`, `[Drop]`, `[Breakdown]`, `[Refrain]`, `[Spoken]`, `[Interlude]`, `[Prelude]`, `[Sample]`, etc. Adding a blank newline between sections yields the best results.
* In some very rare cases, you can also scarcely use Ad libs (vocal embellishments) to your prompts in `(parentheses)`, only when necessary. Examples include `(yeah)`, `(alright)`, `(come on)`, `(whoa)`, etc. Ad libs tend to work best at the end of a line but can also work mid-line. Unlike metatags, ad libs are sung/verbalized.
* In some very rare cases, you can also scarcely use non-lexical vocables, only when necessary. Examples include `la la la`, `na na na`, `sha na na`.

**Example:**
```
[Verse]
I've been tryna call
I've been on my own for long enough
Maybe you can show me how to love, maybe

[Chorus]
I said, ooh, I'm blinded by the lights
No, I can't sleep until I feel your touch
I said, ooh, I'm drowning in the night
Oh, when I'm like this, you're the one I trust
Hey, hey, hey
```

Write the lyrics below after the "Lyrics:" section. You are not allowed to add any narrative or text before or after your response such as "Here's your answer" or "Below is the response".

**Lyrics:**

Table 11: Prompt template for generating song lyrics from lyrics features. Here, {lyrics_feature} indicates placeholder for input lyrics features.

**Task:**
Given a song, you need to conduct a comprehensive stylistic analysis and extract all relevant information about the song's style. This includes identifying characteristics such as instruments used (e.g., guitar, drums, piano, violin, synthesizers, bass, orchestral, solo, acoustic, trumpet, saxophone, flute, cello, harmonica, banjo, accordion, etc.), vocals types (male or female), genres (e.g., rock, pop, pop rock, indie rock, hard rock, metal, heavy metal, r&b, electronic, soul, jazz, country, reggae, classical, hip hop, blues, folk, punk, funk, disco, alternative, grunge, etc.), tempo (slow, moderate, fast), mood (e.g., melancholic, upbeat, aggressive, melodic, sad, happy, excited, nostalgic, mellow, serene, joyful, dark, gothic, etc.), and any other stylistic elements (e.g., dance, party, cinematic, dreamy, energetic, relaxing, anthemic, atmospheric, groovy, etc.) that contribute to the overall vibe, environment, or atmosphere of the song.

**Your response should be structured as follows:**
```
Answer:
<start>
style1, style2, style3, ..., styleN
<end>
```

For example:
```
Answer:
<start>
male vocals, electronic, guitar, piano, energetic, pop rock,
violin, upbeat, dance, synth, sad, soul, trumpet, reggae
<end>
```

Please note that the list of style elements should be comprehensive and cover all relevant aspects of the song's style. Ensure that your response follows strictly to the specified formatting, including the use of angle brackets, commas, and space separating each element written in lowercase. There must not be any narrative or text in the answer (e.g., 'Here's your answer' or 'Below is the response'), only the listed style elements.

{song}

Table 12: Prompt template for extracting song style (e.g. vocal type, instruments) from audio song. Here, {song} indicates placeholder for input audio song.

**Task:**

You are a talented songwriter, music director, and composer. Your task is to compose a {genre} genre song about {topic} with a {mood} mood. Provide the lyrics and style of the song after the "Answer" section. Follow the step-by-step instructions provided in the "Instructions" section and respond using the format given in the "How to Answer" section.

**Instructions:**

1. Before you write the song, you need to plan what you will write about the song, then synthesize the features of the song lyrics mentioned below.
* Subject Matter: Write what the song is about by providing a summary of the story, narrative, central topic, or events discussed in the song lyrics.
* Theme: Identify the main theme or message (emotion, life experience, social commentary, or philosophical concept).
* Target Audience: Define the intended audience (age group, cultural background, or specific interests).
* Narrative: The story and point of view (e.g., first person or second person).
* Character Analysis: Main characters (e.g., Protagonist, Antagonist) and their traits.
* Song Structure: Outline the structure (number of verses, choruses, bridges, intros, and outros) and note any unique elements or deviations.
* Mood: Describe the overall mood (upbeat, melancholic, introspective, etc.).
* Reference: Identify any cultural, social, time, place, or contextual references.
2. Then, you need to write song lyrics that include all the features described in the "Lyrics Features" section. The song should not be long; rather, it should be medium-length. You should also write the lyrics with metatags following the song structure from "Lyrics Features." In some very rare cases, you can also scarcely include ad libs, or non-lexical vocables. Here are the detailed instructions:
* You can add metatags to your lyrics on top of a section in [square brackets] that will create certain styles. Some examples of metatags are [Verse], [Chorus], [Bridge], [Solo], [Outro], [Pre-Chorus], [Bridge], [Hook], [Opening], [Intro], [Instrument], [Build], [Drop], [Breakdown], [Refrain], [Spoken], [Interlude], [Prelude], [Sample], etc. Adding a blank newline between sections yields the best results.
* In some very rare cases, you can also scarcely use Ad libs (vocal embellishments) to your prompts in (parentheses), only when necessary. Examples include (yeah), (alright), (come on), (whoa), etc. Ad libs tend to work best at the end of a line but can also work mid-line. Unlike metatags, ad libs are sung/verbalized.
* In some very rare cases, you can also scarcely use non-lexical vocables, only when necessary. Examples include la la la, na na na, sha na na.
3. Finally, you need to compose the song by synthesizing all relevant and detailed information about the song's style. This includes identifying characteristics such as instruments used (e.g., guitar, drums, piano, violin, synthesizers, bass, orchestral, solo, acoustic, trumpet, saxophone, flute, cello, harmonica, banjo, accordion, etc.), vocals types (female or male), genres (e.g., rock, pop, pop rock, indie rock, hard rock, metal, heavy metal, r&b, electronic, soul, jazz, country, reggae, classical, hip hop, blues, folk, punk, funk, disco, alternative, grunge, etc.), tempo (slow, moderate, fast), mood (e.g., melancholic, upbeat, aggressive, melodic, sad, happy, excited, nostalgic, mellow, serene, joyful, dark, gothic, etc.), and any other stylistic elements (e.g., dance, party, cinematic, dreamy, energetic, relaxing, anthemic, atmospheric, groovy, etc.) that contribute to the overall vibe, environment, or atmosphere of the song. The name of the style must be written in lowercase and separated by commas.
4. Finally, choose a title for the song that best suits its lyrics and style.

Prompt continued on next page...

..continued from previous page

**How to answer:**
You need to provide your answer after the "Answer" section below, while strictly following the format below. Also in your answer, you are not allowed to use any text formatting (bold face, italic, etc.) or narrative (Here's your answer, Answer is below, etc.). Just provide your answer using the below format:

Lyrics Feature:
```
<feature>
* subject_matter: "....."
* theme: "...."
* target_audience: "...."
* narrative: "...."
* character_analysis: "...."
* song_structure: "...."
* mood: "...."
* reference: "...."
</feature>
```

Song Lyrics:
```
<lyrics>
[Verse 1]
In a small town by the sea, where the waves kiss the shore,
Lives a dreamer with a heart, always yearning for more.
With a notebook in his hand, and a vision in his eyes,
He paints the world in colors, beneath the endless skies.

[Pre-Chorus]
Through the struggles and the trials, he keeps his head up high,
With a song within his soul, he knows he'll touch the sky.

....
</lyrics>
```

Song Style:
```
<style>
male vocals, electronic, guitar, piano, energetic, pop rock,
violin, upbeat, dance, synth, sad, soul, trumpet, reggae, ....
</style>
```

Song Title:
```
<title>
Chasing the Dream
</title>
```

**Answer:**

Table 13: Prompt template for generating song lyrics and style from genre, topic and mood. It also generates lyrics feature and song title as by-product. Here, {genre}, {topic}, {mood} indicates placeholders for input genre, topic and mood of the song.

