# OpenReview forum: "SONICS: Synthetic Or Not - Identifying Counterfeit Songs"
_ICLR.cc/2025/Conference — ICLR 2025 Poster_

### Official Review · Reviewer_znTb · 2024-10-28

**Soundness:** 2
**Presentation:** 2
**Contribution:** 3
**Rating:** 3
**Confidence:** 3

**Summary:**

This paper introduces SONICS, a large-scale dataset designed for end-to-end synthetic song detection, addressing the limitations of existing datasets such as lack of musical diversity, short song durations, and absence of fully AI-generated music. They also propose a novel model, SpecTTTra, which excels in capturing long-range temporal dependencies within songs, outperforming current models in terms of speed and memory efficiency. The paper establishes both human and AI benchmarks to advance research in distinguishing AI-generated music, thereby protecting artistic integrity.

**Strengths:**

The paper presents three data acquisition pipelines for fake songs, significantly expanding the dataset for synthetic song detection and providing valuable reference for peers looking to augment this dataset. The paper also introduces a new model for synthetic song detection that achieves better detection results while reducing training costs, offering a new reference method for the field.

**Weaknesses:**

1. The data collection for fake songs is not comprehensive, leading to potential biases. Only songs generated by Suno and Udio were collected, while other AI music generation tools such as SeedMusic and SkyMusic exist in the market.
2. Music generated by AI tools from the same company tends to have a consistent style. The authors need to demonstrate how they can distinguish between real and fake songs, rather than just identifying the music styles of Suno and Udio. Suno and Udio's song styles are concentrated in certain genres that, while present in real-world music, are less common. For instance, if the model achieves an 80% accuracy rate, it might mean that 20% of the ground truth song collection consists of pop and rock styles generated by Suno and Udio, while the remaining 80% are styles rarely generated by these tools. Thus, the model might be performing genre classification rather than accurately distinguishing between real and fake songs.
   I suggest that to reduce this interference and truly identify AI and non-AI differences, rather than stylistic differences, the authors could introduce songs generated by other AI tools, different from the training data, as a test set. For example, if the model is trained to distinguish between real and fake songs using data from Suno and Udio, it could be tested with songs generated by SkyMusic, SeedMusic, and other models. If the model can still identify songs from SkyMusic and SeedMusic as fake, it would demonstrate the model's effectiveness.

3. The authors need to prove that they are identifying real songs versus fake songs, not just differences in audio quality. The dataset includes real songs and AI-generated songs. Real songs are sourced from YouTube, where the sampling rate is generally higher and the audio quality is better. In contrast, AI-generated songs mentioned in the paper have a sampling rate of 16kHz/32kHz, which is of poorer quality, and even if upscaled to match YouTube's quality, there would be significant loss in the high-frequency spectrum.
   The paper does not explore or explain this aspect, nor does it mention whether the audio quality was normalized before training and testing. It is unclear whether the model has truly learned to distinguish AI-generated songs or if it has been judging audio quality.

   Whether it's distinguishing between 16kHz resolution songs and high-resolution songs or differentiating between up-sampled 16kHz songs and original high-resolution songs, it's clear that these are audio quality tasks, not real song vs fake song recognition tasks. Only by reducing all to the same low-quality audio can the bias of audio quality on the model be avoided.

4. The experimental tables in the paper do not indicate which metrics are the best, making it difficult to intuitively compare the effectiveness from the table.
5. There is a lack of Ablation Study to show the model's performance using only Spectral or Temporal information.
6. There is no experimental comparison with other models and methods used in Synthetic Song Detection papers.

**Questions:**

1. The dataset's limited scope to Suno and Udio-generated songs may introduce bias, overlooking other AI music tools like SeedMusic and SkyMusic.
2. The model's high accuracy might reflect genre classification rather than distinguishing real from fake songs due to the focused style of Suno and Udio.
3. The study lacks evidence to confirm the model's ability to identify fake songs independently of audio quality differences.
4. The paper's experimental tables fail to highlight the best-performing metrics, hindering clear comparison.
5. An Ablation Study is missing to assess the model's reliance on Spectral or Temporal information alone.
6. The paper does not compare the model's performance with other methods from the Synthetic Song Detection literature.

---

> ### Author Response · Authors · 2024-11-24
> **Response by Authors to Reviewer znTb**
>
> We sincerely thank the reviewer for their time and valuable feedback on our work. We are especially grateful for their recognition of our contribution as a **“..valuable reference for peers looking to augment this dataset”** and acknowledgment of our method as one **“..that achieves better detection results while reducing training costs, offering a new reference method for the field.”** Additionally, we appreciate the reviewer’s recognition of the contributions of our work as **“good.”** We have addressed all concerns below:
>
> ---
>
> 1.
> > The data collection for fake songs is not comprehensive, leading to potential biases. Only songs generated by Suno and Udio were collected, while other AI music generation tools such as SeedMusic and SkyMusic exist in the market.
>
> We appreciate the reviewer highlighting this important point. While we used two sources, our dataset spans **five** distinct generative models: Suno v2, v3, v3.5, and Udio 32, 133. The reliance on Suno and Udio reflects the nascent stage of the AI-generated song field during our data collection **in June**. At that time, these were the only accessible sources for end-to-end fake song generation. The tools mentioned by the reviewer, **SeedMusic** and **SkyMusic**, were introduced later (**in August and September, respectively**) and could not be included in our dataset.
>
> Additionally, even if these tools had been available earlier, technical constraints would have limited their inclusion. For example:
>
> - **SeedMusic:** This platform offers only a small set of sample songs on its [website](https://team.doubao.com/en/special/seed-music), without public code or a platform for generating additional songs.
> - **SkyMusic (Mureka):** This tool lacks an API unlike Suno-Udio, making bulk song generation or download infeasible.
>
> Furthermore, Given the dynamic nature of the generative AI field, it is not feasible to continuously update dataset and retrain models to incorporate new releases. Therefore, we believe our dataset reflects the state of the field during the defined collection period.
>
> Finally, we believe that our dataset, despite its limitations, lays the groundwork for the novel task of end-to-end synthetic song detection and serves as a foundation for advancing research in song forensics. We also believe it will inspire others to expand and refine this work as the field evolves. Furthermore, given the widespread popularity of the Suno and Udio platforms, our dataset is particularly relevant for practical, real-world applications in fake song detection.
>
> ---

---

> ### Author Response · Authors · 2024-11-24
> **Continuation of Response by Authors to Reviewer znTb**
>
> 2.
> > Music generated by AI tools from the same company tends to have a consistent style.
>
> We thank the reviewer for the valuable concern. Even though we use songs from two sources, there are five distinct algorithms (Suno v2, v3, v3.5, Udio 33, 130), which may share some similarities but exhibit distinct features, as evidenced by the detector's varying performance on them in `Table 4`.
>
>
>
> > The authors need to demonstrate how they can distinguish between real and fake songs, rather than just identifying the music styles of Suno and Udio.
>
> In media forensics, **every generator leaves forensic fingerprints/artifacts specific to its architecture** as mentioned in [1][2]. Moreover, it is very difficult to completely disentangle fake song detection from generator/source detection because fake song detection spans a generative vastness with no defined closed set of patterns. Thus, different generators may produce different types of fake songs. Moreover, it is possible that the detector is identifying the fingerprints of Suno and Udio as part of the fake song detection process, which is not an unexpected behavior. Even with songs from 10, 20, or more sources/companies, the detector still could focus on generator-specific fingerprints. In the audio processing domain, a closely related field to this topic is **"Synthetic Speech Attribution"** [3], which deals with identifying the source/generator of synthetic speech.
>
> It is also important to note that even if a detector is unable to accurately detect songs from an unseen generator, this does not diminish its usefulness. For example, with the recent explosion of fake songs from Suno and Udio, our method could be effectively deployed to detect or flag AI-generated songs on music platforms like Spotify or YouTube Music.
>
> ### Reference:
> - [1] N. Yu, L. Davis, M. Fritz, Attributing Fake Images to GANs: Learning and Analyzing GAN Fingerprints, In ICCV 2019
> - [2] U. A. Çiftçi, İ. Demir, L. Yin , Deepfake source detection in a heart beat, In The Visual Computer, Volume 40, pages 2733–2750, (2024)
> - [3] D. Salvi, P. Bestagini, S. Tubaro, Exploring the Synthetic Speech Attribution Problem
> Through Data-Driven Detectors, In  2022 IEEE International Workshop on Information Forensics and Security (WIFS)
>
>
>
>
> > I suggest that to reduce this interference and truly identify AI and non-AI differences, rather than stylistic differences, the authors could introduce songs generated by other AI tools, different from the training data, as a test set.
>
> Unfortunately, currently we are unable to find any viable sources/datasets for end-to-end fake songs. Tools like SeedMusic and SkyMusic, as mentioned earlier, have limitations, with no feasible way to generate enough samples for proper evaluation (e.g., lack of code/platform or API). Thus, it is not possible for us to measure the detector’s performance beyond Suno and Udio.  .

---

> ### Author Response · Authors · 2024-11-24
> **Continuation of Response by Authors to Reviewer znTb**
>
> > For example, if the model is trained to distinguish between real and fake songs using data from Suno and Udio, it could be tested with songs generated by SkyMusic, SeedMusic, and other models. If the model can still identify songs from SkyMusic and SeedMusic as fake, it would demonstrate the model's effectiveness.
>
> Thank you for your insightful suggestion regarding testing our model with songs generated by unseen generators like SkyMusic and SeedMusic. However, evaluating a detector's ability to handle unseen generators is a specific sub-domain of media forensics research and relates to the ongoing challenge of **"Generalization"** [1, 2, 3]. This task is **different from the primary goal of our paper** that is to address the lack of existing datasets for the end-to-end synthetic song detection task by introducing a novel dataset for this purpose, while also highlighting the importance of long-context modeling for this task, tackling the computational challenges associated with it, and offering human evaluation benchmarks that were lacking in existing works. While generalization to unseen generators is related to our work, **it is beyond the immediate scope of this study**. Nonetheless, we believe our dataset and findings will serve as a valuable foundation for future research aimed at improving generalization capabilities in fake song detection.
>
> That said, we believe the **inability to accurately detect fakes songs from unseen generators does not imply a failure in fake song detection as a whole**, given **detecting fakes songs/media from known generators remains a critical and non-trivial task** within this domain with real-life applications. Specifically, for our case, detecting Suno- and Udio-generated songs is of paramount importance due to the recent surge in fake songs from these sources.
>
> However, due to the relevance of this aspect, we also planned to evaluate our detector on data from SkyMusic and SeedMusic. Unfortunately, **due to the unavailability of code, platforms, or APIs for these generators, we were unable to generate the necessary data for evaluation**.
>
> ### Reference:
>
> - [1] C. Tan, H. Liu, Y. Zhao, S. Wei, G. Gu, P. Liu, Y. Wei, Rethinking the Up-Sampling Operations in CNN-based Generative Network for Generalizable Deepfake Detection. In IEEE/CVF Conference on Computer Vision and Pattern Recognition (CVPR), 2024, pp. 28130-28139
> - [2] Y. Xu, K. Raja, L. Verdoliva, M. Pedersen, Learning Pairwise Interaction for Generalizable DeepFake Detection. In IEEE/CVF Winter Conference on Applications of Computer Vision (WACV) Workshops, 2023, pp. 672-682
> - [3] D. Cozzolino, K. Nagano, L. Thomaz, A. Majumdar, L. Verdoliva, Synthetic Image Detection: Highlights from the IEEE Video and Image Processing Cup 2022 Student Competition. In IEEE Signal Processing Magazine
>
>
>
>
>
> > Suno and Udio's song styles are concentrated in certain genres that, while present in real-world music, are less common. For instance, if the model achieves an 80% accuracy rate, it might mean that 20% of the ground truth song collection consists of pop and rock styles generated by Suno and Udio, while the remaining 80% are styles rarely generated by these tools. Thus, the model might be performing genre classification rather than accurately distinguishing between real and fake songs.
>
> We thank the reviewer for their insightful concern. Genre classification could indeed occur if Suno and Udio songs were limited to specific genres. However, upon analyzing the distribution of song genres, we found that both real and fake songs span all genres, as provided in the figure below. This figure illustrates **significant overlap across all genres between real and fake songs**. Therefore, if the model were attempting to perform **genre classification as a shortcut for detecting fake songs, its results would have deteriorated**. However, **given the strong performance** of our method, we can confidently conclude that **our model is not relying on genre classification**.
>
> * **figure**: https://i.postimg.cc/nLKbcRZN/genre-distribution.png

---

> ### Author Response · Authors · 2024-11-24
> **Continuation of Response by Authors to Reviewer znTb**
>
> ---
>
> 3.
> > The authors need to prove that they are identifying real songs versus fake songs, not just differences in audio quality. The dataset includes real songs and AI-generated songs. Real songs are sourced from YouTube, where the sampling rate is generally higher, and the audio quality is better. In contrast, AI-generated songs mentioned in the paper have a sampling rate of 16kHz/32kHz, which is of poorer quality, and even if upscaled to match YouTube's quality, there would be significant loss in the high-frequency spectrum. The paper does not explore or explain this aspect, nor does it mention whether the audio quality was normalized before training and testing. It is unclear whether the model has truly learned to distinguish AI-generated songs or if it has been judging audio quality.
>
> We thank the reviewer for raising this concern. It was mentioned in the paper (`Section 4.2, Page 8, Line 381-383`) that spectrograms are generated with a sample rate of 16kHz, meaning both real and fake songs are resampled to 16kHz. Thus, it is not possible for the detector to overfit based on audio quality.   To ensure clarity, we have rephrased this in the paper as follows:
>
> *"To train models, we resampled both real and fake songs to 16kHz and generated spectrograms with n_fft = win_length = 2048, hop_length = 512, and n_mels = 128, yielding a 128 × 128 spectrogram for 5 seconds and 128 × 3744 for 120-second audio."*
>
> ---
>
> 4.
> > The experimental tables in the paper do not indicate which metrics are the best, making it difficult to intuitively compare the effectiveness from the table.
>
> We thank the reviewer for raising this valuable concern. To clarify, we have made the table captions more descriptive and highlighted the best-performing variant with bold and underline. Additionally, we have updated the paper (`Page 8, Lines 400–402`) to mention that we use the F1 score as our primary metric.
>
> ---
>
> 5.
> > There is a lack of an Ablation Study to show the model's performance using only Spectral or Temporal information.
>
> We thank the reviewer for this valuable concern. We have added a detailed ablation study in `Section 4.4 with Table 7` (table added below), analyzing the contribution of Spectral and Temporal Tokens/Information. This study demonstrates that both spectral and temporal information are important for fake song detection. Furthermore, they complement each other, as combining them outperforms using either spectral or temporal information alone.
>
> **Table for 5 seconds**
>
> | **Temp. Clip Size (t)** | **# Temp. Tok. (T/t)** | **Spec. Clip Size (f)** | **# Spec. Tok. (F/f)** | **F1** |
> |:------------------------:|:---------------------:|:-----------------------:|:----------------------:|:------:|
> | 3                        | 128                   | -                       | 0                      | 0.76   |
> | 3                        | 128                   | 5                       | 25                     | 0.78   |
> | 3                        | 128                   | 3                       | 42                     | 0.79   |
> | 3                        | 128                   | 1                       | 128                    | 0.80   |
> | -                        | 0                     | 1                       | 128                    | 0.75   |
>
> **Table for 120 seconds**
>
> | **Temp. Clip Size (t)** | **# Temp. Tok. (T/t)** | **Spec. Clip Size (f)** | **# Spec. Tok. (F/f)** | **F1** |
> |:------------------------:|:---------------------:|:-----------------------:|:----------------------:|:------:|
> | 3                        | 1248                  | -                       | 0                      | 0.91   |
> | 3                        | 1248                  | 5                       | 25                     | 0.94   |
> | 3                        | 1248                  | 3                       | 42                     | 0.96   |
> | 3                        | 1248                  | 1                       | 128                    | 0.97   |
> | -                        | 0                     | 1                       | 128                    | 0.92   |

---

> ### Author Response · Authors · 2024-11-24
> **Continuation of Response by Authors to Reviewer znTb**
>
> 6.
> > There is no experimental comparison with other models and methods used in Synthetic Song Detection papers.
>
> We thank the reviewer for the valuable feedback. Given the novel nature of the end-to-end synthetic song detection task, there is currently no existing dataset and, therefore, no existing methods specifically designed for this task. Existing methods in the literature focus on Singing Voice Deepfake Detection (SVDD), which differs from End-to-End Synthetic Song Detection, as outlined in `Table 2`. However, recognizing their relevance, we have compared the methods mentioned in SingFake [1] on our dataset and provided the results in `Appendix, Section A.1` with Table 8 (also added below). Our analysis demonstrates that the SpecTTTra models outperform the compared methods in terms of effectiveness while maintaining efficiency. For example, although the AASIST [2] models deliver promising results for short songs, they are computationally intensive, as reflected by their high time and memory consumption. Additionally, AASIST models fail to process long songs of 120 seconds, resulting in out-of-memory (OOM) errors due to its extreme computational complexity. On the other hand, while ResNet variants exhibit high efficiency, they lack the capacity to detect fake songs as effectively as SpecTTTra, particularly with long-duration songs. In contrast, the SpecTTTra models strike a balance between efficiency and effectiveness. As shown in the Table, SpecTTTra-$\alpha$ achieves the highest F1 score of 0.97 for long-duration songs while maintaining manageable computational requirements. These results establish the superiority of our method in both detecting fake songs and handling longer audio inputs compared to other existing methods.
>
> **Table for 5 Seconds**
>
> | Model              | Speed (A/S) ↑ | FLOPs (G) ↓ | Mem. (GB) ↓ | # Act. (M) ↓ | # Param. (M) ↓ | F1 ↑  |
> |--------------------|---------------|-------------|-------------|--------------|----------------|-------|
> | AASIST            | 55            | 12          | 6           | 96           | 0.3            | 0.91  |
> | ResNet + Spec.     | 354           | 0.6         | 0.1         | 0.8          | 11             | 0.86  |
> | ResNet + LFCC      | 331           | 0.6         | 0.1         | 0.8          | 11             | 0.88  |
> | Wav2Vec2 + AASIST  | 42            | *           | 1.3         | *            | 95             | 0.90  |
> | SpecTTTra-γ        | 154           | 0.7         | 0.1         | 2            | 17             | 0.76  |
> | SpecTTTra-β        | 152           | 1.1         | 0.2         | 2            | 17             | 0.78  |
> | SpecTTTra-α        | 148           | 2.9         | 0.5         | 6            | 17             | 0.80  |
>
>
> **Table for 120 Seconds:**
>
> | Model              | Speed (A/S) ↑ | FLOPs (G) ↓ | Mem. (GB) ↓ | # Act. (M) ↓ | # Param. (M) ↓ | F1 ↑  |
> |--------------------|---------------|-------------|-------------|--------------|----------------|-------|
> | AASIST            | 2             | 295         | OOM         | 2393         | 0.3            | -     |
> | ResNet + Spec.     | 146           | 17.2        | 2.3         | 24           | 11             | 0.90  |
> | ResNet + LFCC      | 144           | 17.2        | 2.3         | 25           | 11             | 0.91  |
> | Wav2Vec2 + AASIST  | 2             | *           | OOM         | *            | 95             | -     |
> | SpecTTTra-γ        | 97            | 10.1        | 1.6         | 20           | 24             | 0.88  |
> | SpecTTTra-β        | 80            | 14.0        | 2.3         | 29           | 21             | 0.92  |
> | SpecTTTra-α        | 47            | 23.7        | 3.9         | 50           | 19             | 0.97  |
>
> It is noteworthy that we were unable to run all the models, particularly the AASIST [2] variants, for 120-second songs due to their tremendous computational complexity, which exceeds our available resources.
>
>
> ### Reference:
> - [1] Y. Zang, Y. Zhang, M. Heydari, Z. Duan, SingFake: Singing Voice Deepfake Detection. In  2024 IEEE International Conference on Acoustics, Speech and Signal Processing (ICASSP)
> - [2]  Jung, Jee-weon, et al. "Aasist: Audio anti-spoofing using integrated spectro-temporal graph attention networks." ICASSP 2022-2022 IEEE international conference on acoustics, speech and signal processing (ICASSP). IEEE, 2022.

---

### Official Review · Reviewer_LAdp · 2024-10-28

**Soundness:** 3
**Presentation:** 3
**Contribution:** 3
**Rating:** 6
**Confidence:** 4

**Summary:**

The paper introduces SONICS (Synthetic Or Not - Identifying Counterfeit Songs), featuring the largest synthetic song dataset to date with over 97,000 songs (4,751 hours), including 49,074 AI-generated songs from Suno and Udio platforms. SONICS includes end-to-end generated content with both music and lyrics, and features longer duration songs (averaging 176 seconds). To effectively detect these synthetic songs, the authors propose SpecTTTra (Spectro-Temporal Tokens Transformer), a novel architecture specifically designed for capturing long-range patterns in music. This new model demonstrates superior efficiency compared to traditional approaches, running 38% faster than ViT while using 26% less memory and achieving better detection performance, particularly with longer audio samples. This work provides essential tools for detecting AI-generated music as these technologies become increasingly sophisticated and widespread.

**Strengths:**

- The paper presents a clear and reproducible workflow for synthetic song generation, with detailed documentation of the prompt engineering process, generation parameters, and data processing steps.
- The generated song dataset can be an important addition to SVS and SVDD community.
- The Spectro-Temporal tokenizer uses a simple but seemingly effective way to capture the long-range information.

**Weaknesses:**

- The paper lacks sample demonstrations of the generated synthetic songs for evaluation. Without access to audio examples, it is difficult for reviewers and readers to assess the quality and diversity of the generated content, as well as validate the claims about different types of synthetic artifacts (e.g., "Karaoke effect"). The authors should provide a representative set of samples across different generation categories (Full Fake, Half Fake, Mostly Fake) and generation platforms (Suno, Udio).
- The naming "SpecTTTra" (Spectro-Temporal Tokens Transformer) is potentially misleading as the architecture only employs a transformer encoder component rather than a full transformer architecture. Furthermore, while the model claims to better capture long-range dependencies, the paper lacks ablation studies to demonstrate this capability. The authors should provide specific experiments showing how the separate temporal and spectral tokenization contributes to capturing long-range patterns in music.
- The performance improvements over baseline models, particularly ConvNeXt, are modest given similar computational costs. The F1 score gain of 1% with 20% speed improvement doesn't strongly justify the introduction of a new architecture. The authors should either provide additional experiments demonstrating clear advantages of SpecTTTra (e.g., ablation studies, analysis of captured patterns), or consider separating the dataset contribution from the model architecture into two papers.
- The quotation mark is used incorrectly.

**Questions:**

- Could the authors provide a representative set of audio samples from different categories (Full Fake, Half Fake, Mostly Fake) and platforms (Suno, Udio)?
- What quality control measures were used to ensure the generated songs are suitable for the dataset?
- Could the authors provide ablation studies demonstrating how the separate temporal and spectral tokenization contributes to capturing long-range patterns?
- What was the rationale behind the specific clip sizes chosen for the three variants (α, β, γ)? And how sensitive is the model performance to these hyperparameters?

---

> ### Author Response · Authors · 2024-11-24
> **Response by Authors to Reviewer LAdp**
>
> We thank the reviewer for taking the time to review our work. We are grateful that the reviewer appreciates our contributions, describing them as **“...an important addition to the SVS and SVDD community”**, **“...a simple but seemingly effective way to capture long-range information"**, and **“...a clear and reproducible workflow for synthetic song generation”**. Additionally, we appreciate the reviewer’s recognition of the contributions and presentation of our work as **“good.”** We have addressed all concerns below:
>
> ---
> 1.
> > The paper lacks sample demonstrations of the generated synthetic songs for evaluation….The authors should provide a representative set of samples across different generation categories (Full Fake, Half Fake, Mostly Fake) and generation platforms (Suno, Udio).
>
> We thank the reviewer for raising this concern. To address it, we have included sample demonstrations for all categories (Full Fake, Half Fake, and Mostly Fake songs) from both Suno and Udio platforms. Additionally, we have added extra samples to showcase different artifacts, such as mechanical voice.
>
> Anonymous link to samples is provided in a **additional comment** below.
>
> ---
>
> 2.
> > The naming "SpecTTTra" (Spectro-Temporal Tokens Transformer) is potentially misleading as the architecture only employs a transformer encoder component rather than a full transformer architecture.
>
> We thank the author for raising this concern. It is valid that we have only used one transformer component. However, many widely recognized models, which also employ a single transformer component, include "transformer" in their names. For example,
> - ViT (Vision Transformer): only uses transformer encoder
> - GPT (Generative Pre-trained Transformer): Only uses transformer decoder.
>
> While it is common in the literature to include "transformer" in model names even when only one component (encoder or decoder) is used, we are open to revisiting the naming. Our intent is not to mislead, and we are willing to consider a more precise name upon discussion if our current choice could cause any confusion or misrepresentation.
>
> > Furthermore, while the model claims to better capture long-range dependencies, the paper lacks ablation studies to demonstrate this capability. The authors should provide specific experiments showing how the separate temporal and spectral tokenization contributes to capturing long-range patterns in music.
>
> We sincerely thank the reviewer for this valuable suggestion. As shown in Table 4 and discussed in `Section 4.3.1 (Lines 401–403)`, our results demonstrate significant performance gains (e.g., 6% for ConvNeXt, 8% for EfficientViT, 10% for ViT, and 17% for SpecTTTra-$\alpha$) when using long songs (120 sec) compared to short songs (5 sec), underscoring the importance of long-context information for enhancing fake song detection.
>
> Additionally, we have included an ablation study (`Section 4.4` with `Table 7`, also below) as per inquiry to clarify the contributions of temporal and spectral tokenization for long context patterns. The study reveals that both play a crucial role: temporal tokens show a 15% gain, and spectral tokens show a 17% gain when using long songs. This confirms that the improvement with increased context size stems from both token types, further supporting the importance of long-range patterns in our approach.
>
> **Table for 5 seconds**
>
> | **Temp. Clip Size (t)** | **# Temp. Tok. (T/t)** | **Spec. Clip Size (f)** | **# Spec. Tok. (F/f)** | **F1** |
> |:------------------------:|:---------------------:|:-----------------------:|:----------------------:|:------:|
> | 3                        | 128                   | -                       | 0                      | 0.76   |
> | 3                        | 128                   | 5                       | 25                     | 0.78   |
> | 3                        | 128                   | 3                       | 42                     | 0.79   |
> | 3                        | 128                   | 1                       | 128                    | 0.80   |
> | -                        | 0                     | 1                       | 128                    | 0.75   |
>
> **Table for 120 seconds**
>
> | **Temp. Clip Size (t)** | **# Temp. Tok. (T/t)** | **Spec. Clip Size (f)** | **# Spec. Tok. (F/f)** | **F1** |
> |:------------------------:|:---------------------:|:-----------------------:|:----------------------:|:------:|
> | 3                        | 1248                  | -                       | 0                      | 0.91   |
> | 3                        | 1248                  | 5                       | 25                     | 0.94   |
> | 3                        | 1248                  | 3                       | 42                     | 0.96   |
> | 3                        | 1248                  | 1                       | 128                    | 0.97   |
> | -                        | 0                     | 1                       | 128                    | 0.92   |
>
> ---

---

> ### Author Response · Authors · 2024-11-24
> **Continuation of Response by Authors to Reviewer LAdp**
>
> 3.
> > The performance improvements over baseline models, particularly ConvNeXt, are modest given similar computational costs. The F1 score gain of 1% with 20% speed improvement doesn't strongly justify the introduction of a new architecture. The authors should either provide additional experiments demonstrating clear advantages of SpecTTTra (e.g., ablation studies, analysis of captured patterns), or consider separating the dataset contribution from the model architecture into two papers.
>
> We thank the reviewer for this insightful comment and appreciate the opportunity to address these points. As highlighted in the `abstract (Lines 26–29)`, our proposed method achieves not only a 20% reduction in time consumption but also a 67% reduction in memory consumption compared to ConvNeXt. Additionally, Table 6 provides further evidence of the significant efficiency gains: SpecTTTra-$\alpha$ achieves a 50% reduction in FLOPs, a 32% reduction in parameters, and a 61% reduction in activation counts compared to ConvNeXt. These improvements represent substantial advancements in computational efficiency.
>
> It is worth noting that previous research papers in the community have been published with comparably modest gains than ours. For instance, FasterViT-3 [1] achieved a 1.31% accuracy improvement over ConvNeXt-B with a 20% faster inference speed, but at the cost of an 18% increase in FLOPs and an 80% increase in parameter count. In contrast, our method delivers both improved performance and enhanced efficiency across all metrics.
>
> The introduction of our architecture is integral to a core aspect of our study: addressing the importance of long-context modeling in fake song detection. Our exploration revealed that existing models are highly inefficient when applied to long-context tasks. This inefficiency motivated the development of a new architecture that balances accuracy with efficiency. Given the demonstrated gains across all metrics and high relevance, we believe the introduction of a new architecture is justified.
>
> ### Reference:
>
> - [1] A. Hatamizadeh, G. Heinrich, H. Yin, A. Tao, J. M. Alvarez, J. Kautz, P. Molchanov, FasterViT: Fast Vision Transformers with Hierarchical Attention. In ICLR 2024
>
> ---
>
> 4.
> > The quotation mark is used incorrectly.
>
> We thank the reviewer for noticing these typos. We have fixed it in our updated manuscript.
>
> ---
>
> 5.
> > What quality control measures were used to ensure the generated songs are suitable for the dataset?
>
> We designed a human-in-the-loop system to ensure that the generated songs were suitable for the dataset. After generating the songs, we relied on human verification to probabilistically sample 1,000 synthetic songs to evaluate potential issues. During our observations, we did not notice any alarming deviations from real songs; however, we found that some samples of Udio contained songs without vocals. This inspired us to use PyAnnotate to filter out songs lacking vocals. Additionally, we implemented text-based filtering to ensure that each synthetic lyric included song meta tags, such as [verse] and [intro], similar to real songs.
>
> ---
>
> 6.
> > What was the rationale behind the specific clip sizes chosen for the three variants (α, β, γ)?
>
> We thank the reviewer for this thoughtful question. The parameters were chosen in a creative manner, with the temporal clip size $t$ kept larger than the spectral clip size $f$. This design accounts for the expectation that long songs contain more temporal information, while the spectral dimension ($F$) remains static regardless of time duration.
>
> > And how sensitive is the model performance to these hyperparameters?
>
> The model's performance is indeed highly sensitive to the parameters $t$ (temporal clip size) and $f$ (spectral clip size), as they directly influence the architecture by controlling the number of temporal and spectral tokens. As demonstrated in Tables 4 and 7, reducing these clip sizes results in performance gains, as it increases the number of respective tokens.
>
> ---

---

### Official Review · Reviewer_m36c · 2024-11-03

**Soundness:** 3
**Presentation:** 3
**Contribution:** 3
**Rating:** 5
**Confidence:** 4

**Summary:**

This paper introduces a dataset of real and "fake" music recordings, where the fake recordings have been generated using commercial music generation systems. An attempt has been made to match some characteristics between the real and fake recordings. In addition, a new transformer architecture is proposed that is aimed at capturing longer-term temporal context in music audio classification. Finally, results on the fake vs. real music classification task are presented.

**Strengths:**

The introduction of a dataset for fake vs. real music classification is a welcome and very timely contribution. Different systems by two major commercial providers are considered and the dataset is separated in terms of different levels of fake (fully, mostly, half), which provides interesting insights - notably the half-fake songs with real lyrics seem to be easier to recognize. In the provided transformer architecture, I believe it makes a lot of sense to have one token per time step rather than several tokens for different frequency patches.

**Weaknesses:**

Unfortunately, the steps of the dataset preparation pipeline do not seem to be evaluated rigorously, which limits the usefulness of the resulting dataset. I am also not very convinced by the usefulness of the proposed spectral tokens.
Overall, because of these and some other issues (see "Questions"), I recommend to reject this paper, although this is a borderline recommendation.

**Questions:**

Major questions/comments:
- Line 81f: "the fake songs in our dataset are free from copyright issues, as they are generated through paid subscriptions that provide a license to use and share the content legally" --- Given that the generation models used are very likely to have been trained on copyrighted data, how can you ensure that the fake songs in your dataset are free from copyright issues? What if generations are identical or similar to existing real music recordings - in particular, when real lyrics and styles are used for generation (cf. "half fake" subset).
- Several steps in the dataset preparation pipeline will probably produce very noisy or incorrect results, but their performance is never evaluated. This is concerning, given that the dataset is the main contribution of this paper. In particular:
  - How well does PyAnnotate work for vocal detection on this data (line 189)?
  - How accurate are the extracted song-styles? This, in particular, I would expect to work rather poorly. Vocal type classification from music audio, for example, is a hard problem.
  - How accurate are the transcribed lyrics?
  - In the cases were generated instead of transcribed lyrics were used: How often do these regurgitate the lyrics from real songs?
- In the proposed SpecTTTra model, I am unconvinced that the proposed spectral clips can provide very helpful information for long songs. The resulting spectral tokens would contain something similar to an average spectrum, which would not be very informative for real vs. fake classification. Indeed, increasing the number $f$ of spectral clips does not improve results in Table 4. However, this was not independently evaluated from the choice of $t$.
- Following up on the previous comment, it seems like results for the SpecTTTra-$\alpha$ model are best, even though this utilizes the proposed architecture the least (low $f$ and $t$).

Minor questions/comments:
- In Table 1, consider adding "Total hours".
- Line 142f: "Meanwhile, long audio classification remains an uncharted area in audio research." --- This statement seems too strong and should be removed. A wide range of music audio classification tasks involve long audio classification, for example: genre classification, global key detection, global tempo estimation, etc.
- Line 211f: How was the list of topics for the full fake generation obtained? The list given in the appendix seems very biased. Is "matt damon vs jimmy kimmel" really a representative example of a topic for a real song?
- Line 288: Why were significantly more MF songs than HF or FF songs included?
- Line 471: What was the musical experience level of the human annotators? Did they have access to spectral analysis tools (which could be helpful for the fake vs. real classification task)?

Additional comments:
- Line 12: "these inventions democratize music creation" --- This is a highly questionable, opinionated slogan that has its place in marketing but not in science. It should be removed.
- Line 68 (and others): Please do not use the bibliography to refer to companies such as Suno or Udio. A company is not a scientific publication. Same, e.g., for wandb (line 392).
- Table 4: Consider clarifying in the caption what the positive and negative classes are (i.e. what is real and what is fake). Furthermore, indicate which algorithms are seen and unseen during training.

---

> ### Author Response · Authors · 2024-11-24
> **Response by Authors to Reviewer m36c**
>
> We sincerely thank the reviewer for taking the time to evaluate our work. We are grateful that the reviewer considers our work to be **“...a welcome and very timely contribution.”** Additionally, we appreciate the reviewer’s recognition of the contributions, soundness, and presentation of our work as **“good.”** Below, we have addressed all the concerns raised by the reviewer.
>
> ---
> ### Major Comments:
>
> 1.
> > Line 81f: "the fake songs in our dataset are free from copyright issues, as they are generated through paid subscriptions that provide a license to use and share the content legally" --- Given that the generation models used are very likely to have been trained on copyrighted data, how can you ensure that the fake songs in your dataset are free from copyright issues?
>
> We understand the gravity of reviewer's concern regarding the copyright issue. Statutory copyright law specifically `17 U.S.C. § 107` [1] states that *“Notwithstanding the provisions of sections 106 and 106A, the fair use of a copyrighted work, including such use by reproduction in copies or phonorecords or by any other means specified by that section, for purposes such as criticism, comment, news reporting, teaching (including multiple copies for classroom use), scholarship, or research, is not an infringement of copyright.”* So, even if the generative models are trained using copyrighted data, we believe our work falls under **fair use** according to the **research** criteria.
>
> Furthermore, using generative models (which likely have been trained on copyrighted data) to generate large-scale datasets is present in the literature and **published in peer-reviewed and widely accepted conferences**. Some examples: LLaVA-Instruct-158K (from GPT4) [2], Gpt4tools (from ChatGPT) [3], Camel (from GPT 3.5 turbo) [4], Baize Dataset (from ChatGPT) [5], Cifake Dataset (uses StableDiffusion to generate fake images) [6], JourneyBench Dataset (uses  GPT-4V and GPT-4o) [7], etc.
>
>
> > What if generations are identical or similar to existing real music recordings - in particular, when real lyrics and styles are used for generation (cf. "half fake" subset).
>
> We thank the reviewer for raising this concern. To address this, we compared all the fake songs with all the real songs in a pairwise manner using the cosine similarity metric with EfficientNetB0 embeddings as representations. Among the top 50 songs with the highest similarity to real songs, we manually inspected each one and found no cases of exact matches. All these songs exhibited variation in either music style, vocals, instrumentation, or other elements, including the "half fake" subset where only lyrics were shared. Given this variation, we believe this work also falls under **fair use policies** as outlined in `17 U.S.C. § 107` [1].
>
> ### Reference:
>
> - [1] https://uscode.house.gov/view.xhtml?path=/prelim@title17&edition=prelim
> - [2] Liu, H., Li, C., Wu, Q., & Lee, Y. J. (2024): Visual instruction tuning. Advances in Neural Information Processing Systems, 36.
> - [3] Yang, R., Song, L., Li, Y., Zhao, S., Ge, Y., Li, X., & Shan, Y. (2024). Gpt4tools: Teaching large language model to use tools via self-instruction. Advances in Neural Information Processing Systems, 36.
> - [4] Li, G., Hammoud, H., Itani, H., Khizbullin, D., & Ghanem, B. (2023). Camel: Communicative agents for" mind" exploration of large language model society. Advances in Neural Information Processing Systems, 36, 51991-52008.
> - [5] Canwen Xu, Daya Guo, Nan Duan, and Julian McAuley. 2023. Baize: An Open-Source Chat Model with Parameter-Efficient Tuning on Self-Chat Data. In Proceedings of the 2023 Conference on Empirical Methods in Natural Language Processing, pages 6268–6278, Singapore. Association for Computational Linguistics.
> - [6] Bird, J. J., & Lotfi, A. (2024). Cifake: Image classification and explainable identification of ai-generated synthetic images. IEEE Access.
> - [7] Wang, Zhecan, et al. "JourneyBench: A Challenging One-Stop Vision-Language Understanding Benchmark of Generated Images.". In NeurIPS 2024.
>
> ---

---

> ### Author Response · Authors · 2024-11-24
> **Continuation of Response by Authors to Reviewer m36c**
>
> 2.
> > Several steps in the dataset preparation pipeline will probably produce very noisy or incorrect results, but their performance is never evaluated. This is concerning, given that the dataset is the main contribution of this paper.
>
> We thank the reviewer for raising this concern. While AI-assisted data generation or annotation can introduce potential errors, this is a widely accepted practice in the research community, as demonstrated by datasets proposed in LLaVa [2] and GLaMM [8]. Given the large scale of such datasets, exhaustive manual verification is not feasible and considered a fair trade-off. However, as per reviewer’s concern we have evaluated our pipeline, as detailed below, which shows that any potential errors have minimal impact on the dataset's overall quality.
>
> > How well does PyAnnotate work for vocal detection on this data (line 189)?
>
> We used PyAnnotate following the methodology in previous work, SingFake [8]. To evaluate its performance, we manually analyzed 50 random samples from our dataset and found that this tool is highly accurate in detecting vocals versus non-vocals. While potential errors might exist, we determined that their impact is negligible. Specifically:
>
> * **False negatives** (classifying songs with no vocals as having vocals): Human evaluators reported that, in their evaluation, they did not find any songs consisting solely of background music without vocals.
> * **False positives** (classifying songs with vocals as non-vocals): The total number of samples classified as non-vocal by PyAnnotate is only 3% of the dataset. Even if we conservatively assume that all of these are false positives (which is not the case based on our evaluation), the overall impact on the dataset remains minimal.
>
> Given these findings, any potential errors introduced by PyAnnotate affect only a negligible portion of the dataset and do not compromise dataset’s integrity.
>
> > How accurate are the extracted song-styles? This, in particular, I would expect to work rather poorly. Vocal type classification from music audio, for example, is a hard problem.
>
> To evaluate the accuracy of the extracted song styles, we manually analyzed 50 randomly selected real songs and compared their AI-generated song styles (genre, instruments, mood, and vocal type) with their actual styles retrieved from internet. Our analysis showed that the generated styles were highly accurate for genre, instruments, and mood. However, we observed occasional inaccuracies in vocal type classification. Despite this limitation, we believe this issue does not significantly affect the goal of our work, which is fake song detection.
>
>
> > How accurate are the transcribed lyrics?
>
> We believe there may have been a misunderstanding, as our pipeline does not involve transcribing lyrics (song → lyrics). Instead, the lyrics in our dataset are generated through large language models (LLMs). Specifically:
>
> * **Full Fake Songs (FF)**: Lyrics are generated using an LLM based on randomly selected topics, genres, and moods.
> * **Mostly Fake Songs (MF)**: Lyric features are extracted from real lyrics using an LLM and used to generate new lyrics using an LLM.
> * **Half Fake Songs (HF):** Lyrics are taken from Genius Lyrics Dataset [9].
>
> Since lyrics in our dataset are generated using large language models (LLMs), the process eliminates any potential risk of transcription errors. Hope this clears any misunderstanding.
>
>
> > In the cases were generated instead of transcribed lyrics were used: How often do these regurgitate the lyrics from real songs?
>
> To evaluate whether generated lyrics regurgitate real lyrics, we conducted a pairwise comparison of all fake lyrics with all real lyrics using cosine similarity based on SentenceTransformer embeddings (`all-MiniLM-L6-v2`). Among the top 50 pairs with the highest similarity, we manually inspected each case and found no instances of exact matches. The generated lyrics consistently exhibited meaningful differences in length, phrasing, expression, story, and structure.
>
> Additionally, we randomly analyzed 50 "mostly fake" songs to assess the broader quality of the generated lyrics. While thematic overlaps (e.g., subject matter, mood) were observed, significant differences emerged in story flow, mood depth, and stylistic choices, confirming that the LLM-generated lyrics are original and not regurgitated from real songs.
>
>
> ### Reference:
> - [8] H. Rasheed, M. Maaz, S. Shaji, A. Shaker, S. Khan, H. Cholakkal, R. M. Anwer, E. Xing, MH Yang, F. S. Khan, GLaMM: Pixel Grounding Large Multimodal Model. In CVPR 2024
> - [9] Carlos G. D. C. J. Genius song lyrics with language information. https://www.kaggle.com/datasets/carlosgdcj/genius-song-lyrics-with-language-information, 2023. Accessed: 2024-06-05.

---

> ### Author Response · Authors · 2024-11-24
> **Continuation of Response by Authors to Reviewer m36c**
>
> 3.
> > In the proposed SpecTTTra model, I am unconvinced that the proposed spectral clips can provide very helpful information for long songs. The resulting spectral tokens would contain something similar to an average spectrum, which would not be very informative for real vs. fake classification.
>
> We thank the reviewer for raising this valuable concern. To address this, we conducted an ablation study (`Section 4.4 and Table 7`) that demonstrates both spectral and temporal information contribute meaningfully to fake song detection. While each can independently classify real vs. fake songs, combining them yields better performance, showcasing their complementary nature.
>
> **Table for 5 seconds**
>
> | **Temp. Clip Size (t)** | **# Temp. Tok. (T/t)** | **Spec. Clip Size (f)** | **# Spec. Tok. (F/f)** | **F1** |
> |:------------------------:|:---------------------:|:-----------------------:|:----------------------:|:------:|
> | 3                        | 128                   | -                       | 0                      | 0.76   |
> | 3                        | 128                   | 5                       | 25                     | 0.78   |
> | 3                        | 128                   | 3                       | 42                     | 0.79   |
> | 3                        | 128                   | 1                       | 128                    | 0.80   |
> | -                        | 0                     | 1                       | 128                    | 0.75   |
>
> **Table for 120 seconds**
>
> | **Temp. Clip Size (t)** | **# Temp. Tok. (T/t)** | **Spec. Clip Size (f)** | **# Spec. Tok. (F/f)** | **F1** |
> |:------------------------:|:---------------------:|:-----------------------:|:----------------------:|:------:|
> | 3                        | 1248                  | -                       | 0                      | 0.91   |
> | 3                        | 1248                  | 5                       | 25                     | 0.94   |
> | 3                        | 1248                  | 3                       | 42                     | 0.96   |
> | 3                        | 1248                  | 1                       | 128                    | 0.97   |
> | -                        | 0                     | 1                       | 128                    | 0.92   |
>
>
> Interestingly, for long-duration songs, spectral tokens performed better than temporal tokens, likely due to the smaller number of spectral tokens (~128 vs. ~1248 temporal tokens), which reduces token redundancy and aids in extracting useful information. Moreover, increasing song duration improved performance for both spectral and temporal tokens, supporting our claim of importance of long-context information in fake song detection.
>
>
> > Indeed, increasing the number $f$ of spectral clips does not improve results in `Table 4`. However, this was not independently evaluated from the choice of $t$. Following up on the previous comment, it seems like results for the SpecTTTra-$\alpha$ model are best, even though this utilizes the proposed architecture the least (low $f$ and $t$).
>
>
> We believe there may have been a misunderstanding. In our paper, $f$ and $t$ represent the size of spectral and temporal clips, not the number of clips. As shown in `Eq. 2` of the paper, where $T$ and $F$ are the temporal and spectral dimensions of the input spectrogram:
>
> $N_{\psi} = N_{spectral} + N_{temporal} = \left(\frac{F}{f}\right) + \left(\frac{T}{t}\right)$
>
> Increasing $f$ and $t$ reduces the number of spectral and temporal tokens/clips, which explains the observed drop in performance with larger clip sizes. From the recently added `Table 7`, this relationship is further clarified by showing the impact of varying $f$ while keeping $t$ constant. Specifically, reducing $f$ (i.e., increasing the number of spectral tokens/clips) improves performance, underscoring the importance of spectral information for both short and long duration songs. Therefore, SpecTTTra-$\alpha$ model achieving best performance is very expected due to its lowest $t$ and $f$ value.

---

> ### Author Response · Authors · 2024-11-24
> **Continuation of Response by Authors to Reviewer m36c**
>
> ### Minor Comments:
>
> 1.
> > In Table 1, consider adding "Total hours".
>
> Thank you for your thoughtful suggestion. We have incorporated the “# Total Hours” column into Table 1 as recommended. This addition highlights that our dataset comprises a total of 4,751 hours, which is significantly greater than other datasets (15–182x more hours). This difference is primarily due to the inclusion of long-duration songs in our dataset.
>
> ---
>
> 2.
> > Line 142f: "Meanwhile, long audio classification remains an uncharted area in audio research." --- This statement seems too strong and should be removed. A wide range of music audio classification tasks involve long audio classification, for example: genre classification, global key detection, global tempo estimation, etc.
>
> We thank the reviewer for raising this concern. As per suggestion, we've toned down the statement as such *"Meanwhile, long audio classification remains a relatively less explored area in audio research"*.
>
> ---
>
>
> 3.
> > Line 211f: How was the list of topics for the full fake generation obtained? The list given in the appendix seems very biased. Is "matt damon vs jimmy kimmel" really a representative example of a topic for a real song?
>
>
> We thank the reviewer for this concern.The topic list was generated in a semi-automatic fashion, where a carefully manually curated list of topics was passed to an LLM to create similar topics with a broad range. While we acknowledge that some of the topics might seem biased, we account for real-world scenarios where any user can create songs for any topic. For example, users can create parody songs on a variety of topics, and there seems to be a tendency among social media to upload songs on these types of biased issues relevant to pop culture. To ensure our model performs well in this real-world scenarios for fake song detection, we incorporate fine-grained topics inspired from pop culture and current affairs along with topics more aligned to real songs.
>
> ---
>
> 4.
> > Line 288: Why were significantly more MF songs than HF or FF songs included?
>
> We thank the reviewer for this valuable question. The distribution of song types was designed to ensure balance and minimize potential biases during training:
>
> * **HF Songs:** These contain only Suno-generated algorithms. Including too many HF samples would create an imbalance between Suno and Udio sources, potentially skewing model performance.
> * **FF Songs:** Since both lyrics and song styles in FF songs are generated from entirely random topics, genres, and moods, their characteristics may deviate significantly from real songs. A larger proportion of FF songs could lead to model overfitting on fake songs having potential larger impact on overall performance.
> * **MF Songs:** These are closer to real songs in terms of lyrics and song style, making them less likely to induce overfitting. Therefore, we included a higher proportion of MF songs to avoid potential overfitting.
>
> ---
>
> 5.
> > Line 471: What was the musical experience level of the human annotators? Did they have access to spectral analysis tools (which could be helpful for the fake vs. real classification task)?
>
> We thank the reviewer for this concern. The human annotators involved in our study were everyday music listeners without formal training or expertise in music theory or composition. Additionally, they did not utilize any spectral analysis tools during the annotation process.
>
> Our primary motivation for human benchmarking was to assess the extent to which generative models can conceal their artifacts from typical human perception. Given that the largest audience for synthetic music comprises general consumers rather than professional musicians, we aimed to ensure that the annotator distribution accurately reflected this real-world demographic.
>
> ---
>
> ### Additional Comments
> 1.
> > Line 12: "these inventions democratize music creation" --- This is a highly questionable, opinionated slogan that has its place in marketing but not in science. It should be removed.
>
> We thank the reviewer for the suggestion and we've updated the manuscript to reflect the feedback.
>
> ---
>
> 2.
> > Line 68 (and others): Please do not use the bibliography to refer to companies such as Suno or Udio. A company is not a scientific publication. Same, e.g., for wandb (line 392).
>
> We thank the reviewer for the suggestion. We have updated the manuscript to reflect the feedback.
>
> ---
>
> 3.
> > Table 4: Consider clarifying in the caption what the positive and negative classes are (i.e. what is real and what is fake). Furthermore, indicate which algorithms are seen and unseen during training.
>
> We thank the reviewer for the suggestion. We have updated the manuscript to reflect the feedback.

---

### Official Review · Reviewer_hEps · 2024-11-03

**Soundness:** 3
**Presentation:** 3
**Contribution:** 3
**Rating:** 6
**Confidence:** 4

**Summary:**

This paper presented a dataset for detecting AI generated music from well-known AI generated music services, Suno and Udio. The authors basically configured two group of music which are fake and real songs. For real songs, they retrieved 48,090 real songs from YouTube and for fake songs, they generated music from Suno and Udio. Finally, they trained a model for detecting whether the song is real or fake. To do this, SpecTTTra model is proposed. In this model, the authors proposed to use spectral and temporal patches (full-size in one dimension) to construct spectral and temporal tokens for their transformer-based classification model. The results showed that the proposed model is highly effective in detection. Also, the authors compared the model performance with Human performance.

**Strengths:**

Overall, the paper is well-written and easy-to-read, and the details of the proposed method, experimental design, the dataset, and the experimental results are well addressed. I think the following researchers will be easily follow and use the proposed dataset, and the models as the authors described them well.

**Weaknesses:**

It is good to see some research paper in the direction of detecting AI generated content in the domain of music. However, I think the paper lacks of analysis part. The scores that's shown in result tables seem to prove the effectiveness of the proposed method, however, without such in-depth analysis, the proposed dataset and the proposed method might not be useful in the future if the algorithms are updated. Therefore, I think having some analysis section would be beneficial for further future researches. (I think leaving some comments that is related to more high-level explanation on why the proposed model is detecting the AI generated content well, e.g. vocal characteristics, temporal/timbral characteristics, etc) At least, some additional case study on Appendix would be nice. For example, If some real song is detected as a fake, then try to give some analysis on why it's been detected as fake or vice versa.

**Questions:**

I think the title of the paper and the term real and fake are a bit too generalized terms.
For example, the proposed dataset is about detection on Suno and Udio generated contents. (not counterfeit songs, maybe synthetic is okay)
I think the authors can carefully suggest a better title.

---

> ### Author Response · Authors · 2024-11-24
> **Response by Authors to Reviewer hEps**
>
> We greatly appreciate the reviewer’s efforts in thoroughly reviewing our work and providing valuable feedback. We are also sincerely grateful for the reviewer’s recognition of our contributions, particularly noting, **“...details of the proposed method, experimental design, the dataset, and the experimental results are well addressed,”** and **“It is good to see some research paper in the direction of detecting AI-generated content in the domain of music.”** Furthermore, we value the reviewer’s acknowledgment of the soundness, presentation, and overall quality of our work as **“good.”** Below, we have addressed all the concerns raised by the reviewer.
>
> ---
>
> > However, I think the paper lacks an analysis part…. Therefore, I think having some analysis section would be beneficial for further future researches.
>
> We thank the reviewer for this valuable suggestion. We have added a detailed analysis section in `Appendix A.2` to enhance the interpretability of the results and provide a deeper understanding of our model's behavior. Below is a short overview of our analysis:
>
> - **True Negatives:** Correctly classified real songs exhibit unique human-like characteristics such as unpredictability, dynamic variation, non-standard pitch variations, intricate rhythmic complexity, and expressive techniques like melismatic phrasing, tempo changes, and improvisation.
> - **True Positives:** Correctly classified fake songs show typical AI-generated artifacts, such as mechanical vocal qualities, limited pitch variability, predictable rhythmic structures, and reduced emotive expressiveness.
> - **False Negatives:** Misclassified fake songs lack typical AI artifacts and mimic human-like unpredictability and nuanced variation, often incorporating features like spoken interludes or conversational segments that resemble genuine music.
> - **False Positives:** Misclassified real songs share characteristics often found in AI-generated music, such as unclear vocals, reduced rhythmic complexity, and absence of the unpredictable variations.
> - **Imperceptible Artifacts:** Some cases likely exhibit subtle, imperceptible, or inaudible artifacts [1], analogous to "invisible artifacts" [2] in synthetic images, making detection particularly challenging.
>
> ### Reference:
> - [1] Barrington, Sarah, et al. "Single and multi-speaker cloned voice detection: from perceptual to learned features." 2023 IEEE International Workshop on Information Forensics and Security (WIFS). IEEE, 2023
> - [2] Chhabra, Saheb, et al. "Low quality deepfake detection via unseen artifacts." IEEE Transactions on Artificial Intelligence (2023).

---

> ### Author Response · Authors · 2024-11-24
> **Continuation of Response by Authors to Reviewer hEps**
>
> > I think the title of the paper and the term real and fake are a bit too generalized terms. For example, the proposed dataset is about detection on Suno and Udio generated contents. (not counterfeit songs, maybe synthetic is okay) I think the authors can carefully suggest a better title.
>
> Here, the term **"real and fake songs"** aligns with terminology used interchangeably with **"genuine and synthetic"** in existing literature, such as FSD [4], SingFake [5], and CtrSVDD [6]. The word **"counterfeit"** used in our title is considered a synonym for "fake." Therefore, we believe a change may not be necessary. Nonetheless, we remain open to exploring alternative terms through further discussion and feedback.
>
> ### Reference
>
> - [4] Y. Xie, J. Zhou, X. Lu, Z. Jiang, Y. Yang, H. Cheng, FSD: An Initial Chinese Dataset for Fake Song Detection. In 2024 IEEE International Conference on Acoustics, Speech and Signal Processing (ICASSP)
> - [5] Y. Zang, Y. Zhang, M. Heydari, Z. Duan, SingFake: Singing Voice Deepfake Detection. In  2024 IEEE International Conference on Acoustics, Speech and Signal Processing (ICASSP)
> - [6] Y. Zang, J. Shi, Y. Zhang, R. Yamamoto, J. Han, Y. Tang, S. Xu, W. Zhao, J. Guo, T. Toda, Z. Duan, CtrSVDD: A Benchmark Dataset and Baseline Analysis for Controlled Singing Voice Deepfake Detection. In Interspeech 2024

---

> > ### Comment · Reviewer_hEps · 2024-11-26
> > **I will keep my score**
> >
> > I think the proposed dataset and methods would only be useful for Suno and Udio's certain version models. Therefore, it might be better to describe Suno3.5 / Udio130 explicitly even in the Title. (The rest concern has been resolved)

---

> > > ### Author Response · Authors · 2024-11-27
> > >
> > > We sincerely thank the reviewer for taking time to re-evaluate our work. We are glad that **our response has addressed most of the concerns**, leaving only one point for further clarification. Below, we address this remaining concern:
> > >
> > > > I think the proposed dataset and methods would only be useful for Suno and Udio’s certain version models.
> > >
> > > - **We believe our method will be useful beyond specific versions of Suno and Udio.**
> > >   - As highlighted in Table 4 of our paper, our proposed method demonstrates strong performance on algorithms from both Suno and Udio that were unseen during training. Specifically, SpecTTTra-α achieves **96% sensitivity on Udio 33**, **99% on Suno v3**, and **78% on Suno v2**, all of which were not part of the training set. These results indicate robust detection capabilities and suggest that the method has the potential to generalize to other algorithms from the same sources.
> > >   - Moreover, our method **highlights the importance of long-context patterns for fake song detection** across algorithms from two separate sources (e.g., Suno and Udio), as shown in Table 4. Thus, we believe it will be useful for other algorithms as well.
> > >   - Furthermore, the proposed method, SpecTTTra, demonstrates **exceptional efficiency**, particularly for long-context audio/song analysis. This efficiency is independent of the specific datasets or algorithms involved. Since the proposed architecture focuses on leveraging long-range patterns, its design inherently supports applicability to future and diverse algorithmic variations.
> > >
> > > - **We also believe our dataset will be useful beyond Suno and Udio.** As no existing dataset supports end-to-end fake song detection, we strongly believe that our proposed dataset will serve as a foundational stepping stone for advancing research in song forensics. By fostering future research, it can help researchers tackle challenges such as generalization (detecting songs generated by unseen detectors).
> > >
> > > ---
> > >
> > > > Therefore, it might be better to describe Suno3.5 / Udio130 explicitly even in the title.
> > >
> > > We appreciate the reviewer’s suggestion regarding the title and we remain open to making it more specific (e.g., explicitly mentioning Suno and Udio) if the reviewer strongly prefers this adjustment. However, we have observed that generalized titles are commonly used in media forensics research, even when the focus is on specific datasets or algorithms. For instance:
> > >    - *“SINGFAKE: Singing Voice Deepfake Detection”* [1] mostly uses the SoftVC-VITS algorithm/model without including any unseen algorithms in the test data.
> > >    - *“FSD: An Initial Chinese Dataset for Fake Song Detection”* [2] focuses on five specific algorithms, with one of them serving as an unseen algorithm in the test data.
> > >    - *“Detection of GAN-Generated Fake Images”* [3] also follows generalized titling conventions despite using only the CycleGAN generator/algorithm for both training and testing.
> > >
> > > In contrast, our proposed dataset includes five algorithms, with two of them serving as unseen algorithms in the test data. Considering these cases, **we feel our current title aligns with common practices in the field**. Nevertheless, if the reviewer believes it is necessary, we are more than willing to revise the title to explicitly mention Udio and Suno.
> > >
> > >
> > > ### Reference:
> > > - [1] Y. Zang, Y. Zhang, M. Heydari, Z. Duan, SingFake: Singing Voice Deepfake Detection. In 2024 IEEE International Conference on Acoustics, Speech and Signal Processing (ICASSP)
> > > - [2] Y. Xie, J. Zhou, X. Lu, Z. Jiang, Y. Yang, H. Cheng, FSD: An Initial Chinese Dataset for Fake Song Detection. In 2024 IEEE International Conference on Acoustics, Speech and Signal Processing (ICASSP)
> > > - [3] Marra, Francesco, et al. "Detection of gan-generated fake images over social networks." 2018 IEEE conference on multimedia information processing and retrieval (MIPR). IEEE, 2018.

---

### Author Response · Authors · 2024-11-30
**Official Comment on Generalization (Part 2 of 2)**

## Generalization Test Results

| **Len. (sec)** | **Model**         	| **Seed Music**      	| **Sky Music**       	|
|----------------|-----------------------|-------------------------|-------------------------|
| 5          	| AASIST           	| 0.65                	| 0.56                	|
|            	| ResNet + Spec.   	| 0.52                	| 0.48                	|
|            	| ResNet + LFCC    	| 0.54                	| 0.51                	|
|            	| Wav2Vec2 + AASIST	| 0.69                	| 0.53                	|
|            	| ConvNeXt         	| 0.28                	| 0.21                	|
|            	| ViT              	| 0.68                	| 0.57                	|
|            	| EfficientViT     	| 0.67                	| 0.54                	|
|            	| SpecTTTra-γ      	| 0.51                	| 0.53                	|
|            	| SpecTTTra-β      	| 0.53                	| 0.56                	|
|            	| SpecTTTra-α      	| 0.55                	| 0.57                	|
| 120        	| AASIST           	| -                   	| -                   	|
|            	| ResNet + Spec.   	| 0.53                	| 0.35                	|
|            	| ResNet + LFCC    	| 0.56                	| 0.40                	|
|            	| Wav2Vec2 + AASIST	| -                   	| -                   	|
|            	| ConvNeXt         	| 0.36                	| 0.19                	|
|            	| ViT              	| 0.64                	| 0.53                	|
|            	| EfficientViT     	| 0.71                	| 0.55                	|
|            	| SpecTTTra-γ      	| **_0.80_**          	| **_0.60_**          	|
|            	| SpecTTTra-β      	| 0.53                	| 0.56                	|
|            	| SpecTTTra-α      	| 0.56                	| 0.54                	|

---

### Author Response · Authors · 2024-11-30
**Official Comment on Generalization (Part 1 of 2)**

We appreciate the reviewers' concerns regarding generalization and aim to address them in this section. However, we would first like to reiterate that the **primary objective of this study was to introduce the first dataset specifically designed for detecting synthetic songs generated in an end-to-end manner**. A secondary objective was to emphasize the critical role of long-range context in synthetic song detection. In doing so, we also highlighted the computational challenges posed by existing methods, which motivated us to propose an efficient alternative with improved performance. Finally, we provided a human benchmark that has been absent in existing fake song detection literature.

Although generalization is not the primary focus of our research, we recognize its importance for understanding the potential of our proposed dataset and method. To evaluate this, we tested all methods trained on our SONICS dataset on songs generated by **SkyMusic** and **SeedMusic**, platforms distinct from those used for dataset creation. Due to the lack of an API for SkyMusic and the absence of a platform for SeedMusic, we were only able to manually collect 394 songs from SkyMusic and 36 from SeedMusic. While this sample size is limited, it provides a preliminary indication of the dataset’s and models’ generalization capabilities.

## Insights from Generalization Tests
The results, summarized in the `table in Part 2 of this discussion`, reveal several key findings:

- **Performance Shift**: Generalization remains a significant challenge, consistent with trends observed in media forensics literature [1, 2, 3]. Specifically, ConvNeXt, which excelled on the primary dataset, struggled significantly on external songs, dropping to the lowest rank across both durations. This underscores the difficulty of adapting baseline CNN models to unseen data sources.

- **SpecTTTra’s Robustness**: Our proposed model, SpecTTTra-$\gamma$, achieved the best F1 scores—**80%** for SeedMusic and **60%** for SkyMusic. However, larger variants of SpecTTTra (e.g., $\beta$ and $\alpha$) showed greater susceptibility to overfitting, though they still outperformed ConvNeXt in generalization tests.

- **Transformer Superiority**: Models incorporating transformers (e.g., SpecTTTra, ViT, EfficientViT) demonstrated greater resilience compared to CNN-based methods (ConvNeXt, ResNet), indicating their potential in addressing the broader generalization challenge.


## Implications and Future Directions
The score drop observed when detecting synthetic data from out-of-distribution (OOD) generators aligns with patterns documented in prior literature [1, 2, 3], which often require sophisticated generalization strategies beyond the scope of our research. Nevertheless, **the ability of some models to achieve promising results on OOD generators without any sophisticated generalization strategies (e.g., [1], [2], [3]) highlights the potential of our dataset to address generalization challenges across platforms beyond Suno and Udio**, providing a foundation for future research on generalization in song forensics. It is worth noting that, to ensure the reliability of our results, we also tested all methods with randomly initialized weights (i.e., without training). None of the models achieved a score above 0.51.

### Reference
- [1] Utkarsh Ojha, Yuheng Li, Yong Jae Lee, Towards Universal Fake Image Detectors that Generalize Across Generative Models. In CVPR 2023
- [2] David C. Epstein, Ishan Jain, Oliver Wang, Richard Zhang, Online Detection of AI-Generated Images. In ICCV 2023
- [3] Zhiyuan Yan, Yuhao Luo, Siwei Lyu, Qingshan Liu, Baoyuan Wu, Transcending Forgery Specificity with Latent Space Augmentation for Generalizable Deepfake Detection. In CVPR 2024

---

### Author Response · Authors · 2024-12-04
**Final Overview (Part 2 of 2)**

### Responses
* **Using two sources (Suno and Udio) – Potential Bias**: Our dataset utilizes only Suno and Udio sources (covering five generative models) because these were the only accessible tools for end-to-end fake song generation at the time of dataset creation. Newer tools, such as SeedMusic and SkyMusic, were introduced later and also have technical constraints, such as the lack of APIs and platforms unlike Suno and Udio. Moreover, given the dynamic nature of the field, where keep updating the dataset is infeasible due to the constant emergence of new models, our dataset captures the state of the field during the defined collection period.

* **Generalization**: We referenced prior studies to highlight that generators leave specific artifacts in synthetic samples. Even with an increased number of generators, models are likely to focus on these generator-specific artifacts. Addressing this issue requires advanced method, as noted in the literature, which is a separate research domain, beyond our scope. Nonetheless, we tested the generalizability of all methods using out-of-distribution (OOD) generators (SkyMusic and SeedMusic) and observed the expected performance drop. However, despite the absence of advanced generalization techniques, some methods achieved promising performance. Specifically, SpecTTTra outperformed other methods, demonstrating the dataset’s potential to extend beyond Suno and Udio and fostering further research in generalization for song forensics. Note: Due to lack of API and Platform, we were only able to collect few samples from publicly available demo songs on their websites.

* **Efficiency and Performance**: The introduction of SpecTTTra is highly relevant to our dataset as we provide long audio samples and existing approaches become computationally heavy while processing them. Specifically, compared to ConvNeXt (best performing amongst previous approaches), we achieved 1% improvement on the F1 Score, while being 20% faster and using 67% less memory.

* **Existing Works**: Given the novelty of end-to-end fake song detection, there is no existing dataset thus no direct method for this task. However, we compared our work with the closest related methods, Singing Voice Deepfake Detection (SVDD), and our analysis shows that SpecTTTra outperforms these models in both performance and efficiency.

* **Contribution of Spectral and Temporal Tokens**: We conducted an ablation study (`Section 4.4 and Table 7`) that demonstrates both spectral and temporal information contribute meaningfully for fake song detection. While each can independently classify real vs. fake songs, combining them yields better performance across short and long songs, showcasing their complementary nature.

* **Potential copyright issue**: The concern was raised as the generative models used to create the dataset may have been trained on copyrighted music. However, our dataset qualifies as fair use under ``17 U.S.C. § 107`` for research purposes. Additionally, using generative models trained on copyrighted data to create datasets is common, and relevant papers have been published in peer-reviewed conferences. Moreover, pairwise similarity checks verified that no real songs have been copied (generated).

* **Possible overfitting on Song Genre**: We analyzed and found that real and fake songs span all genres with significant overlap (figure provided in discussion). Given this overlap and the model’s strong performance, it is clear that the model does not rely on genre classification to detect fake songs.

* **Possible overfitting on Audio Quality (Sample Rate)**: We resample all the real and fake songs to `16 kHz` thus it is not possible to overfit on audio sample rate. We further clarify this in our paper.

* **Dataset integrity**: We evaluated the steps in our dataset creation pipeline to investigate potential issues and their impact. PyAnnotate showed high accuracy for vocal detection, with negligible impact from rare misclassifications. Furthermore, manual analysis confirmed the high accuracy of AI-extracted song styles, with minor issues in vocal type classification. Moreover, as the lyrics are generated using LLMs, there can be no transcription error as no transcription is being done, while pairwise similarity checks verified that no real lyrics have been regurgitated. These evaluations ensure the dataset's integrity while showing minimal impact from any potential issues.

* **Performance drop with increasing $t$ and $f$** : Increasing $t$ (temporal clip size) and $f$ (spectral clip size) reduces the number of temporal and spectral tokens, as indicated in `Eq. 2`. As a result, performance drops with larger $t$ and $f$.

* **Lack of Samples**: We have provided representative samples across all requested categories (Full Fake, Half Fake, and Mostly Fake) and generation platforms (Suno and Udio) in an anonymous link in the comments.

---

### Author Response · Authors · 2024-12-04
**Final Overview (Part 1 of 2)**

We thank the reviewers for their insightful feedback. We are pleased that all of the reviewers found our contribution **good**. We have addressed all the concerns raised by the reviewers. Specifically, two reviewers, `hEps` and `LAdp`, have already acknowledged that all their concerns have been resolved, while the remaining two reviewers have not yet responded. Below, we summarize our contributions and provide responses to the major concerns:

### Contributions
* Introduced the first dataset specifically designed for detecting end-to-end synthetic songs.
* Provided a human benchmark, which is lacking in existing works, alongside traditional AI benchmarks.
* Emphasized the critical role of long-range context in the detection of synthetic songs and highlighted the computational challenges of existing approaches while processing long audios.
* Proposed an efficient model to process long audio samples, which also provides better performance than existing approaches.

---

### Meta-Review · Area_Chair_PewW · 2024-12-20

**Metareview:**

This paper presented a dataset and a system for detecting AI generated music, using songs from YouTube and fake songs generated from Suno and Udio. The detection system SpecTTTra uses spectral and temporal patches with a transformer-based classification model.
In the evaluation, the test sets include unseen algorithms and unseen singers, showing a reduced performance on OOD data.
Considering that the primary contribution of the paper is providing the first dataset for detecting AI generated songs, this seems as valid accomplishment on its own. The objective of creating a robust detection system is only partly achieved as generalization remains a problem, especially considering later experiments with SkyMusic and SeedMusic.

**Additional Comments On Reviewer Discussion:**

Reviewers raised concerns about possible bias in the dataset, issues of generalization of the model, and several specific technical questions about the representation and detection model parameters. Authors haver responded to all questions. Reviewers m36c and znTb did not follow up and did not respond to the answers provided in the rebuttal. According to my evaluation most of their concerns were adequately addressed.
As mentioned in metareview above, higher weight in ranking the paper is put on the contribution of dataset construction and the proposed pipeline. Robustness of the detection system still remains a major challenge. In terms of the dataset itself, the issue of bias to Suno and Udio also remains a valid concern. Overall I would consider the contributino of data, methods and evaluations to be above threshold, sufficient to merit publication with the purpose to stir more research in this field.

---

### Decision · Program_Chairs · 2025-01-22

Accept (Poster)